# Local genetic context shapes the function of a gene regulatory network

**Anna Nagy-Staron[1]\*, Kathrin Tomasek[1], Caroline Caruso Carter[1], Elisabeth Sonnleitner[2], Bor Kavčič[1], Tiago Paixão[1†], Calin C Guet[1]\***

[1]Institute of Science and Technology Austria, Klosterneuburg, Austria; [2]Department of Microbiology, Immunobiology and Genetics, Max F. Perutz Laboratories, Center Of Molecular Biology, University of Vienna, Vienna, Austria

**Abstract** Gene expression levels are influenced by multiple coexisting molecular mechanisms. Some of these interactions such as those of transcription factors and promoters have been studied extensively. However, predicting phenotypes of gene regulatory networks (GRNs) remains a major challenge. Here, we use a well-defined synthetic GRN to study in *Escherichia coli* how network phenotypes depend on local genetic context, i.e. the genetic neighborhood of a transcription factor and its relative position. We show that one GRN with fixed topology can display not only quantitatively but also qualitatively different phenotypes, depending solely on the local genetic context of its components. Transcriptional read-through is the main molecular mechanism that places one transcriptional unit (TU) within two separate regulons without the need for complex regulatory sequences. We propose that relative order of individual TUs, with its potential for combinatorial complexity, plays an important role in shaping phenotypes of GRNs.

**\*For correspondence:**
anna.staron@gmail.com (AN-S);
calin@ist.ac.at (CCG)

**Present address:** [†]Instituto Gulbenkian de Ciência, Oeiras, Portugal

**Competing interests:** The authors declare that no competing interests exist.

## Introduction

Changes in regulatory connections between individual transcriptional units (TUs) or, in other words, the rewiring of gene regulatory networks (GRNs), is a major genetic mechanism underlying phenotypic diversity (*Shubin et al., 2009*; *Wagner and Lynch, 2010*; *Wray, 2007*). A lot of effort has been put into understanding how mutations in transcription factors and their DNA binding sites within promoter regions influence GRN behavior, plasticity, and evolution (*Babu et al., 2004*; *Balaji et al., 2007*; *Chen et al., 2012*; *Igler et al., 2018*; *Isalan et al., 2008*; *Nocedal et al., 2017*). However, we are still unable to predict GRN phenotypes from first principles (*Browning and Busby, 2016*).

Genes and the genetic elements that regulate them, promoters, are arranged in a linear manner on chromosomes. Thus they are embedded into a larger genetic context, represented by the changing genetic background of the rest of the genomic sequence or by their specific physical location on the chromosome. The genetic context in which GRNs find themselves in, influences and modulates the way these respond to the environment and interact with other GRNs and it also shapes the interactions within the GRN itself (*Cardinale and Arkin, 2012*; *Chan et al., 2005*; *Steinrueck and Guet, 2017*; *Tas et al., 2021*; *Wu and Rao, 2010*). In bacteria, gene expression levels are thought to be determined by RNA polymerase recognizing promoter sequences and subsequently initiating transcription, which is the key step at which a large part of transcriptional regulation takes place (*Browning and Busby, 2004*). However, context effects resulting from occupying a particular location within the genome can significantly alter expression levels (*Junier, 2014*; *Lagomarsino et al., 2015*; *Scholz et al., 2019*). Distance to the origin of replication influences transcription levels due to gene dosage effects, the presence of transcriptionally active and silent regions, as well as spatial and temporal variation in DNA superhelicity, while collisions between DNA replication and transcription influence gene expression levels differently on leading and lagging strands (*Beckwith et al., 1966*; *Bryant et al., 2014*; *Mirkin et al., 2006*; *Sobetzko et al., 2012*; *Vora et al., 2009*). At a local

scale, transcriptional interference, transcription-coupled DNA supercoiling, presence of *cis*-antisense RNA, as well as transcriptional read-through, all link together the expression of neighboring TUs (*Cambray et al., 2013*; *Georg and Hess, 2011*; *Liu and Wang, 1987*; *Reynolds et al., 1992*; *Shearwin et al., 2005*; *Wu and Fang, 2003*). Within operons, number, length, and order of genes can all affect gene expression (*Jacob and Monod, 1961*; *Lim et al., 2011*; *Zipser, 1969*). All of these factors that can individually modulate gene expression vary simultaneously across the genome, with potential for significant combinatorial effects (*Meyer et al., 2018*; *Scholz et al., 2019*).

While these multiple local context-dependent mechanisms are known to modulate gene expression, the qualitative phenotype of a GRN is often thought to be defined solely by the network topology and the gene expression levels of GRN components (*Babu et al., 2004*; *Mangan et al., 2003*; *Payne and Wagner, 2015*), and thus determined simply by the promoter sequences, independent of the physical location of the genes. One of the reasons for this assumption is the belief that *cis*-regulatory changes are less pleiotropic than changes to the protein itself (*Prud'homme et al., 2006*), although some have questioned this (*Stern and Orgogozo, 2008*). However, other non-coding genetic factors such as transcriptional read-through or supercoiling have the potential to change gene expression with the same pleiotropic freedom as *cis*-regulatory changes. Here, we ask how the immediate local genetic context outside of individual TUs of a GRN can alter both the qualitative and quantitative phenotype of a network, and how many phenotypes are accessible for this particular GRN, while the network topology per se remains unchanged. In order to keep the number of genetic interactions to a minimum, we chose to study a synthetic GRN. This tractable system allows for a simplified description of more complex naturally occurring GRNs, where a large number of inherently complex interactions make such a question very difficult to answer experimentally (*Mukherji and van Oudenaarden, 2009*; *Wolf and Arkin, 2003*). Here, we shuffle individual TUs (understood here as the unit formed by: the mRNA coding sequence, the promoter driving its expression and the transcriptional terminator marking the end of the transcribed sequence) of a GRN. In doing so, we alter solely the *local genetic context,* while keeping the actual interactions (topology) within the GRN unchanged and thus the number of interactions to a tractable minimum. We then define the *phenotype of the GRN* as the levels of gene expression measured across four different environments, defined by the presence or absence of two different chemical inducers that alter the binding state of two different well-characterized transcription factors. Qualitative phenotypes are here based on a set of binary output values for each input state, therefore defining different logical operators (e.g. NOR, ON, OFF) (for details see *Threshold for assigning a phenotype to individual GRNs*). Quantitative phenotypes are defined as a set of four expression values varying continuously within one particular behavior. In this way, we systematically explore the space of possible phenotypes of the GRN and thereby we can disentangle the effects of local genetic context from multiple other factors that can affect gene expression levels.

## Results

Our GRN (*Figure 1A*) is composed of the genes coding for three of the best characterized repressors: LacI, TetR, and lambda CI (abbreviations used throughout the text: L, T, and C, respectively), and the promoters they control, P$_{lac}$, P$_{tet}$, and P$_R$. The three repressor genes are transcriptionally interconnected into a GRN, with LacI repressing both *tetR* and its own expression, and TetR repressing expression of *cI*. The controlled promoters are synthetic variants of the P$_L$ promoter of phage lambda with two *tet* or *lac* operator sites located in the direct vicinity of the −35 and −10 promoter elements (*Lutz and Bujard, 1997*). The binding state of TetR and LacI changes in the presence of inducers: anhydrotetracycline (aTc) and isopropyl β-D-thiogalactopyranoside (IPTG), respectively. A *yfp* gene, expressed from a CI-controlled P$_R$ promoter, serves as output. The *yfp* gene is located separately from the rest of the genetic circuit at a transcriptionally insulated locus on the chromosome (*attB* site of phage P21). In our synthetic system, each individual promoter and transcription factor gene it controls are separated from the neighboring one by a strong transcriptional terminator T1 of the *rrnB* gene (*Orosz et al., 1991*), forming an individual TU (*Figure 1B*). We chose T1 as one of the strongest transcriptional terminators in *E. coli* to transcriptionally insulate individual TUs from one another (*Cambray et al., 2013*).

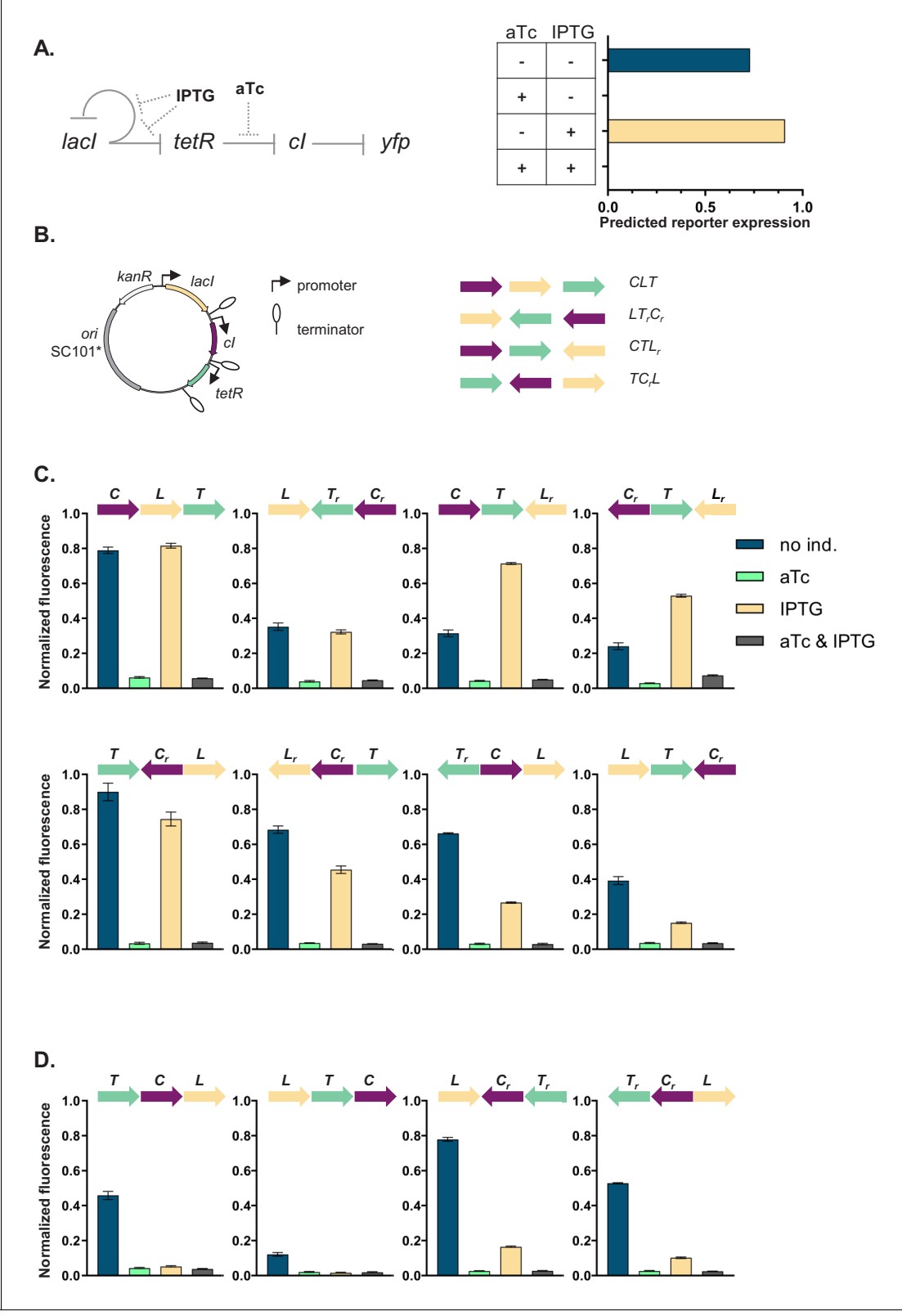

**Figure 1.** Architecture and phenotypes of the gene regulatory network (GRN). (**A**) Diagram of interactions between the three independent transcriptional units (TUs) encoding for the repressors, *lacI*, *tetR*, and *cI*, their respective inducers isopropyl β-D-thiogalactopyranoside (IPTG) and anhydrotetracycline (aTc), and the promoters they control, with *yfp* as the GRN output (left). Phenotype of this GRN as predicted by our mathematical model (right). (**B**) Genetic architecture of TU permutations of GRN plasmid (left). Cartoon of TU permutations (right). Abbreviations used throughout the

*Figure 1 continued*

text: C stands for *cI*, L for *lacI*, and T for *tetR*. Letter *r* denotes reverse orientation. (**C** and **D**) Fluorescence of cells carrying a representative subset of different TU permutations of the GRN plasmid. A binary output value (On or Off) was assigned to each environment which thus defines a logical operation: NOT (aTc) in panel (**C**) and NOR in panel (**D**). Graphs show means and error bars standard deviations for three independent biological replicates.

The online version of this article includes the following figure supplement(s) for figure 1:

**Figure supplement 1.** A model of the mechanistic basis of gene expression, including transcriptional read-through.
**Figure supplement 2.** Gene regulatory networks (GRNs) in which phenotype is dependent only on changes in relative transcriptional unit (TU) order and orientation.
**Figure supplement 3.** Gene regulatory networks (GRNs) in which the influence of plasmid-encoded genetic elements cannot be ruled out.
**Figure supplement 4.** Influence of plasmid genetic elements on gene expression levels.
**Figure supplement 5.** Distribution of phenotypes depending on the threshold applied to define ON and OFF states.

## Phenotype of GRN depends on local genetic context despite identical topology

We asked whether the local genetic context can influence the phenotype of this GRN. First, we developed a simple mathematical model of the mechanistic basis of gene expression for this specific network topology (*Figure 1—figure supplement 1A*). This model tracks the concentrations of all three repressors as their respective promoter activities are influenced by their known specific network interactions. The predicted phenotype of our GRN will depend on the presence of aTc, but not on the presence of IPTG (*Figure 1A*). We wanted to test if this phenotype is independent of the relative TU order and orientation, and we aimed to build plasmids with all possible 48 relative TU order permutations with fixed positions (*Riordan, 2003*) such that: (i) every TU can occupy any of the three positions, (ii) every TU is present only once, (iii) both forward and reverse orientations are possible, and (iv) network topology stays the same (*Figure 1B*). To facilitate comparisons among all GRNs we used a threshold on the expression of the YFP output for assigning a binary output value to each environment and so defined a phenotype the GRN can achieve (for thresholds used to assign a particular phenotype to individual networks, see *Supplementary material*).

The phenotype of strains carrying the resulting 37 plasmids (multiple attempts to clone eleven of the TU order permutations failed, see *Supplemental material*) varied widely both quantitatively and qualitatively (*Figure 1C,D*, *Figure 1—figure supplement 2*, and *Figure 1—figure supplement 3*). More than half (20) of the tested GRN permutations showed a phenotype which was qualitatively different than what was predicted *ab initio*. We also observed multiple quantitative differences in expression levels within one class of logical phenotypes (e.g. permutations CLT, CTL$_r$, T$_r$CL, and LT$_r$C$_r$; *Figure 1C*).

## GRN phenotype is influenced by local genetic context independently of the replicon

We then asked how and why the changes in relative order of individual TUs affect the phenotype of the GRN, although network topology and individual genetic components of the GRN (i.e. individual TUs) remain unchanged. In order to disentangle the specific interactions between network components we focused on six GRNs which differ in relative TU order but not in gene orientation. These six GRNs show two qualitatively different phenotypes (NOR and NOT [aTc]; *Figure 2A*, upper panel). In four out of six strains (LCT, LTC, TCL, and TLC), induction with IPTG shifted *yfp* expression levels to the OFF state. These population-level findings are also observed at the single cell level (*Figure 2A*, lower panel). We tested to what extent our observation from plasmid-based TUs apply to chromosomally located GRNs by integrating three networks with varying TU order (CTL, LCT, and TLC) at a transcriptionally insulated chromosomal locus (*attB* site of phage HK022). In line with the plasmid-based GRNs, these strains also showed a dependency of phenotype on relative TU order, demonstrating that this is not an effect related to plasmid localization of our GRN (*Figure 2B*). When *lacI*, *tetR*, and *cI* are integrated at separate, transcriptionally insulated loci on the bacterial chromosome, the network phenotype is identical with the one predicted *ab initio* from its topology, confirming that it is the transcription of neighboring genes that changes the network's phenotype (*Figure 2C*).

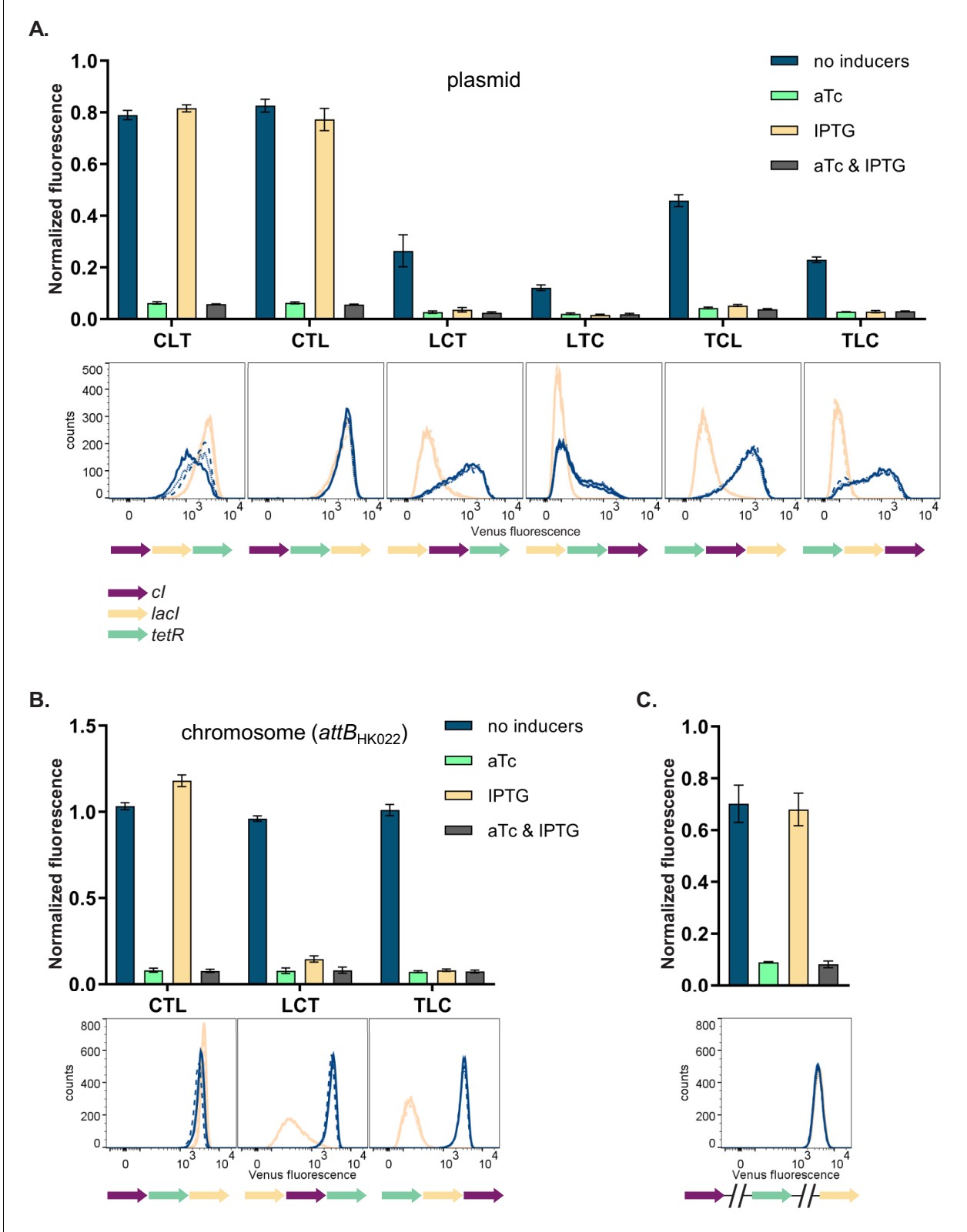

**Figure 2.** Changes in relative transcriptional unit (TU) order lead to qualitative changes of phenotype. Fluorescence of cells carrying six different TU permutations of the gene regulatory network (GRN) on a plasmid (**A**), three GRN variants integrated on the chromosome at the phage HK022 attachment site (**B**), and with each of the repressor genes integrated at separate chromosomal loci (**C**). Graphs show population level fluorescence measurements of strains exposed to: no inducer, anhydrotetracycline (aTc), isopropyl β-D-thiogalactopyranoside (IPTG), or a combination of aTc and

*Figure 2 continued on next page*

*Figure 2 continued*

IPTG (as indicated). Graph shows mean and standard deviations for three independent biological replicates. Flow cytometry histograms of cell fluorescence show 10,000 gated events, corresponding to YFP expressed in a given strain grown without (blue) and with IPTG (yellow). For each strain and condition, three biological replicates are shown. Relative TU order of the three repressors is shown under the respective graphs.

## Differences in cI expression lead to phenotypes that depend on relative TU order

To elucidate the molecular basis of the observed phenotypic variability, we first asked whether the relative TU order-dependent differences in phenotypes can be traced back to changes in levels of *cI* gene expression. We isolated total RNA from strains differing in relative position of the *cI* gene (CLT and TLC) grown in the absence or presence of IPTG, and quantified *cI* transcript levels using RT-qPCR. *cI* expression after IPTG induction in strain TLC was over 10-fold higher than in strain CLT (*Figure 3A*), suggesting that the differences in *yfp* fluorescence were indeed due to differences in *cI* expression. In order to corroborate our findings at the mRNA level with protein expression levels, we replaced the *cI* gene in strains CLT and TLC with *yfp,* and confirmed relative gene order effects on *yfp* expression directly (*Figure 3B*).

Expression of the *cI* gene in our GRN is controlled by TetR (*Figure 1A*). Thus differences in *cI* expression levels between the different relative TU order variants can be due to (i) global changes in gene expression of $P_{tet}$ controlled genes, which propagate to changes in expression of *cI*; or (ii) local effects, such as transcriptional read-through or changes in supercoiling levels. To distinguish between these two possibilities, we measured the activity of the promoter driving *cI* expression by supplying $P_{tet}$-*cfp* in trans on a second plasmid. In all six strains with different relative TU order, $P_{tet}$ activity was strongly induced with aTc, while no $P_{tet}$ activation was observed after IPTG induction (*Figure 3C*). This indicates that *cI* expression observed after IPTG induction is not due to global removal of TetR repression, but rather due to a local effect on *cI* gene expression. Furthermore, levels of TetR-dependent repression do not depend on relative TU order. This local effect depends on gene expression from the *lac* promoter in an IPTG-dependent manner (*Figure 3D*).

## Transcriptional read-through is the molecular mechanism underlying context-dependent GRN phenotype

We hypothesized that transcriptional read-through would be consistent with the context-dependent effects we measured. We observed that in the cases when the *cI* gene is at the second or third position of the GRN (*Figure 1B*), transcriptional read-through from upstream TUs (*tetR* and/or *lacI*) may be enough to transcribe *cI* despite TetR-dependent repression, and in turn shut down $P_R$ activity in response to IPTG (see also *Supplemental material* for effects of transcriptional read-through into *tetR* and *lacI*). We asked whether the potential for transcriptional read-through, together with the knowledge about the individual genetic components of this network, is enough to unambiguously predict the phenotype of the GRN permutation variants we built. For this purpose, we added the effects of transcriptional read-through to our mathematical model such that promoter activities were now influenced not only by network interactions but also by the activity of neighboring genes (*Figure 1—figure supplement 1B*).

To facilitate the analysis of numerous GRNs we divided the 48 possible TU permutations into 24 pairs that differ only in orientation with respect to the plasmid backbone (*Figure 1—figure supplement 2* and *Figure 1—figure supplement 3*). 26 networks/13 pairs showed the same phenotype in both orientations (*Figure 1—figure supplement 2*), eight networks/four pairs showed different phenotype in each orientation (*Figure 1—figure supplement 3A*), for three networks no corresponding pair was cloned (*Figure 1—figure supplement 3B*) and eight networks/four pairs were not cloned.

For networks showing the same phenotype in both orientations we assumed there is no significant influence of plasmid backbone elements (*Figure 1—figure supplement 2*). Here, the model including transcriptional read-through agreed for 20 networks/10 pairs (*Figure 1—figure supplement 2A*) and did not agree with six networks/three pairs (*Figure 1—figure supplement 2B*). A null-model that did not account for transcriptional read-through failed to predict the observed differences in phenotypes.

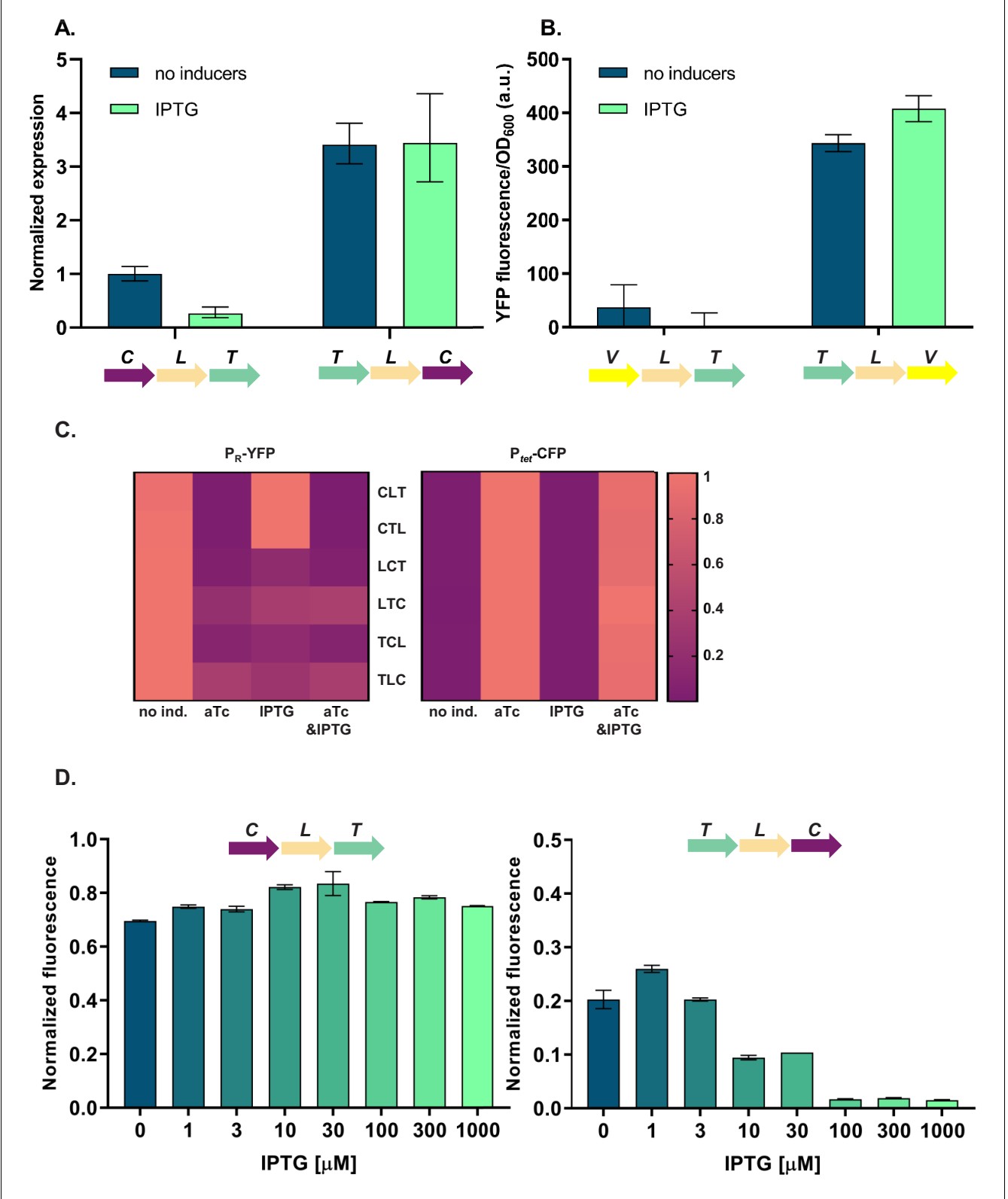

**Figure 3.** Differences in *cl* expression lead to transcriptional unit (TU) order dependent phenotypes. (A) RT-qPCR analysis of *cl* expression. RT-qPCR was performed, using *cl*-specific primers. The induction ratios were calculated relative to the uninduced strain CLT. (B) YFP levels measured in strains VLT and TLV which carry a *yfp* reporter (V) under control of P$_{tet}$ and differ only in relative TU order. (C) Heatmaps show P$_R$ promoter activity in six strains carrying plasmids differing in relative TU order (left) and activity of P$_{tet}$ present in trans in the same strains on a second plasmid (right). (D) Relative TU

*Figure 3 continued on next page*

*Figure 3 continued*

order effects depend on expression from the *lac* promoter in isopropyl β-D-thiogalactopyranoside (IPTG)-dependent manner. Strains CLT and TLC were grown in the presence of different concentrations of IPTG. (**B–D**) Strains were exposed to anhydrotetracycline (aTc), IPTG, or a combination of aTc and IPTG (as indicated). For reasons of clarity, in (**C**) the highest expression level for each strain was individually normalized. Graphs show means and error bars standard deviations for three independent biological replicates.

Our experimental approach to test the transcriptional read-through hypothesis is based on the premise that transcriptional read-through does not depend on a functional promoter of a downstream gene, in contrast to supercoiling- or RNAP concentration-dependent effects. Therefore, if GRN behavior is due to transcriptional read-through, mutating the $P_{tet}$ promoter should not affect the responsiveness to IPTG. We tested this prediction by introducing two point mutations into the −10 element of $P_{tet}$ in a number of different strains (*Figure 4A* and *Figure 4—figure supplement 1*). These two point mutations render $P_{tet}$ inactive and therefore prevent transcription from this promoter (*Figure 4—figure supplement 2*). The phenotype in the absence of any inducers and with only IPTG was identical to the phenotype in the original strains, thus confirming that *cI* expression is initiated at an upstream promoter (*Figure 4B* and *Figure 4—figure supplement 1*).

If the mechanism behind different phenotypes is transcriptional read-through, change of the terminator strength should lead to a change in phenotype. We have chosen network TLC in which our model predicted that a change of terminator strength will lead to alter the phenotype. In this GRN we exchanged the T1 terminator preceding *cI* to either a stronger double T1T2 terminator of the *rrnB* locus, or to the weaker T*crp* and T*tonB* terminators (*Cambray et al., 2013*). Change from T1 to T1T2 changes the phenotype of the network to the one predicted by the model and observed when single transcription factors are incorporated in separate loci on the chromosome (*Figure 5A*). Change from T1 to either T*crp* or T*tonB* leads to a completely OFF phenotype, as expected when transcriptional read-through through weaker terminators leads to expression of *cI* in all four conditions.

Read-through transcripts, i.e. transcripts of more than one gene, can be detected by northern blotting. We isolated total RNA from strains with T1, T*crp*, and T*tonB* grown in the absence or presence of IPTG, and visualized transcripts on northern blot (*Figure 5—figure supplement 1*). We expected to detect *lacI* transcripts in all strains, and in case of transcriptional read-through, longer transcripts encompassing both *lacI* and *cI*. No read-through transcript starting in *tetR* was expected, since the T1 terminator separating *tetR* from *lacI* carries a RNaseE recognition site in its stem (*Apirion and Miczak, 1993*; *Szeberényi et al., 1984*), which makes it impossible to distinguish between read-through transcripts and transcripts originating from individual promoters. The same is true for the strain harboring T1 terminators only, in which we expected to see only single gene transcripts.

We detected read-through transcripts encompassing both *lacI* and *cI* in strains carrying T*crp* and T*tonB*, thus directly demonstrating transcriptional read-through in these two strains. We also detected fragments encompassing only the *cI* gene. This may suggest RNA processing at a cryptic RNase site, or additional effects, such as dislodgement of repressor by RNA polymerase passing through the terminator (*Palmer et al., 2011*). To rule out the emergence of unpredicted promoters, we fused the junctions between *lacI* and *cI*, encompassing T1, T*crp*, and T*tonB* terminators in front of YFP, but did not detect any significant increase in fluorescence (*Figure 5—figure supplement 2*).

In order to check whether a combination of two terminators would act additively on stopping transcriptional read-through, we inserted the weaker *crp* terminator in front of T1 terminator in a different GRN, LCT, resulting in network $L^{crpT1}CT$. This network showed an intermediate phenotype, consistent with predicted decrease in *cI* expression (*Figure 5B*).

Taken together, these results strongly support that transcriptional read-through is the molecular mechanism underlying the relative TU dependent phenotypes we observe, and thus the different logic phenotypes our GRN can achieve.

## Interplay of several molecular mechanisms shapes GRN phenotype

Eight networks (four pairs) showed different phenotypes in each orientation and in all cases one phenotype from the pair was supported by our model while the other was not. This suggests that there was a significant influence of the plasmid backbone elements. Just like the individual TUs

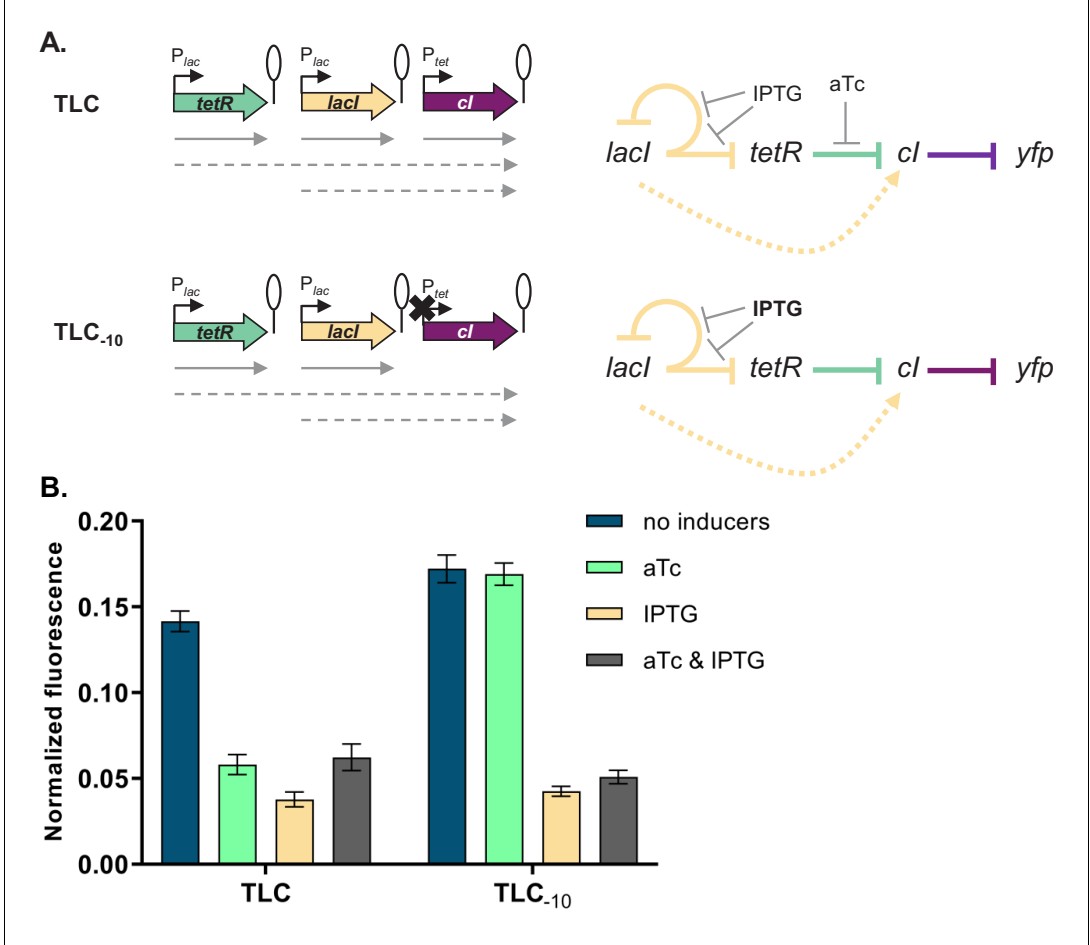

**Figure 4.** Phenotype of strain TLC can be explained by transcriptional read-through. (**A**) Genetic architecture of plasmid fragments encoding three repressors in strains TLC and TLC$_{-10}$ (carrying mutations in the −10 promoter element of P$_{tet}$). Promoters are marked as bent arrows, terminators are represented by vertical bars and a circle, monocistronic transcripts are represented by solid arrows, predicted read-through transcripts by dashed arrows (left). Interaction diagrams within the two gene regulatory networks (GRNs). Solid lines represent interactions between transcription factors and the promoters they control, dashed line represents effects resulting from local genetic context (right). (**B**) Fluorescence of cells carrying TLC plasmid with either P$_{tet}$ or P$_{tet}$ carrying a mutation in the −10 promoter element (TLC$_{-10}$) grown in the presence or absence of anhydrotetracycline (aTc) and isopropyl β-D-thiogalactopyranoside (IPTG). We expected that if GRN behavior is due to transcriptional read-through, mutating the P$_{tet}$ promoter should not affect the responsiveness to IPTG. If, on the other hand, expression of *cl* was driven only by P$_{tet}$, mutating the −10 promoter element should lead to an ALL ON phenotype. Lack of repression in strains with P$_{tet-10}$ variant after aTc induction further confirms that this promoter variant is inactive. Graph shows means and error bars standard deviations for three independent biological replicates.

The online version of this article includes the following figure supplement(s) for figure 4:

**Figure supplement 1.** Phenotype of strains LCT, T$_r$C$_r$L$_r$, TCL, L$_r$C$_r$T$_r$, TLC, and C$_r$L$_r$T$_r$ can be explained by transcriptional read-through.

**Figure supplement 2.** Point mutations in −10 promoter element render P$_{tet}$ inactive.

constituting our GRN, TUs located on the plasmid backbone (namely the kanamycin resistance gene *kanR* and *repA* in the plasmid origin of replication) also have the potential to influence expression of neighboring TUs, and hence the GRN phenotype. To rule out transcriptional read-through from the *repA* gene located at the plasmid origin of replication, we cloned a promoterless *yfp* gene downstream of *repA*; however, we did not detect any significant change in fluorescence (*Figure 1—figure supplement 4A*). We noticed that there is a slight change in repression of P$_{tet}$ depending on the relative orientation to the plasmid backbone (*Figure 1—figure supplement 4B*). Since supercoiling can influence gene expression, especially of plasmid-located genes, we also expect supercoiling–mediated effects to modulate expression and thus influence phenotypes of our GRN (*Sobetzko, 2016*; *Yeung et al., 2017*).

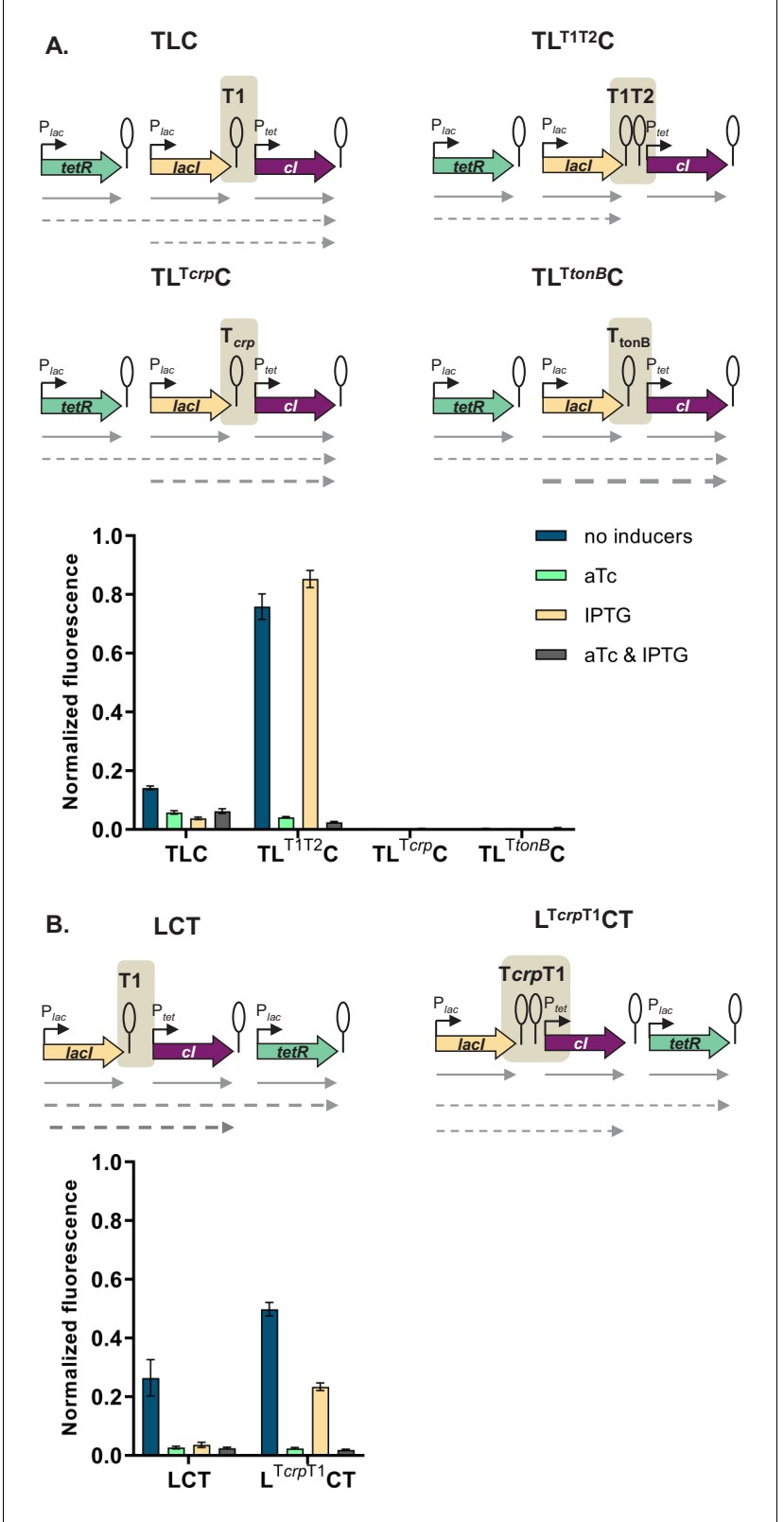

**Figure 5.** Change of terminator leads to qualitative change in phenotype. (**A**) Genetic architecture of plasmid fragments encoding three repressors in strains TLC with different terminators: T1, T1T2, T*crp*, or T*tonB*, preceding *cI*, and fluorescence of cells carrying these plasmids grown in the presence or absence of anhydrotetracycline (aTc) and isopropyl β-D-thiogalactopyranoside (IPTG). (**B**) Genetic architecture of plasmid fragments encoding three

*Figure 5 continued on next page*

*Figure 5 continued*

repressors in strains LCT with either T1 or a double T*crp*-T1 terminator, and fluorescence of cells carrying these plasmids grown in the presence or absence of aTc and IPTG. (**A** and **B**) Promoters are marked as bent arrows, terminators are represented by vertical bars and a circle, operators as rectangles, monocistronic transcripts are represented by solid arrows, predicted read-through transcripts by dashed arrows. Increasing line thickness corresponds to increasing amount of transcript. Graph shows means and error bars standard deviations for three independent biological replicates.

The online version of this article includes the following figure supplement(s) for figure 5:

**Figure supplement 1.** Northern blot assay shows read-through transcript in strains carrying T*crp* and T*tonB*.

**Figure supplement 2.** Terminator containing DNA fragments between repressor genes show no cryptic promoter activity.

## Local genetic context modulates regulation of lac promoter

Our simplified synthetic system allowed us to observe the effects of local genetic context created in a systematic manner by arranging transcription factor genes next to one another on a plasmid (*Figure 2A*) or the chromosome (*Figure 2B*). We also decided to test our findings in a native regulatory network of *E. coli* composed of one transcription factor – LacI, and the promoter it controls, P*lac*, and asked whether native local genetic context has the potential to modulate its phenotype. Using a Δ*lacI* Δ*lacZYA* genetic background we inserted the *lac* promoter driving *yfp* expression in a transcriptionally insulated locus (phage λ attachment site, *attB*) and the *lac* repressor, *lacI*, into one of three loci on the right replichore (*Figure 6A*). Subsequently, we measured gene expression from the *lac* promoter, P*lac*, in these three strains by monitoring *yfp* fluorescence levels after treatment with increasing IPTG concentrations. The shape of the induction curve was considered to be the network's phenotype. It should be stressed that the *lacI* gene (under control of its native promoter) was inserted only into non-coding chromosomal regions, shortly after an endogenous terminator. The loci for insertion were chosen such that genes downstream of the terminator were non-essential and in the same orientation as the genes upstream. Chosen terminators were of different strengths: strong, middle, and weak (D. Toledo Aparicio, M. Lagator, and A. Nagy-Staron, personal communication, September 2019) and were located in close vicinity (39'–43' on MG1655 chromosome) to avoid gene dosage and transcription factor – promoter distance effects (*Block et al., 2012*; *Kuhlman and Cox, 2012*). Moreover, the growth medium (and hence cell growth rate) was chosen such as to further minimize gene dosage effects (*Block et al., 2012*). To assess whether genomic location affected response to IPTG, we measured *yfp* fluorescence reporting on P*lac* expression levels at several points along the IPTG concentration gradient. If *lacI* is inserted after a weaker terminator, expression from *lac* promoter is lower for a range of IPTG concentrations as compared to strain where *lacI* is inserted after a strong terminator (*Figure 6B*). We conducted an analysis of variance (ANOVA) to compare the effect of genomic localization on P*lac* activity. There was a significant effect for five IPTG concentrations tested, and post hoc comparisons using the Tukey test were performed (*Figure 6—figure supplement 1*). We also directly assessed the amount of *lacI* transcript in these three strains using RT-qPCR and saw differences in expression levels consistent with the observed P*lac* induction curves (*Figure 6C*).

In this minimal network consisting of a transcription factor and the promoter it controls, the network phenotype is indeed modulated only by the endogenous local genetic context, likely due to varying levels of transcriptional read-through into the *lacI* gene. To verify this, we performed PCR on cDNA from the three strains, using primers spanning the intergenic regions upstream of *lacI* (*Figure 6D*). In all three cases, a DNA band corresponding to the amplification of the cDNA spanning the intergenic region was obtained, confirming read-through transcription into *lacI* from the upstream gene (no band was obtained with RNA as template). This corroborates our findings that genetic context of network elements can modulate network phenotype and that any kind of chromosomal rearrangement has the potential to alter network output.

## Discussion

By comprehensively shuffling the relative TU order in a synthetic GRN, we show that local genetic context can significantly influence the phenotype of GRNs both quantitatively and qualitatively, and

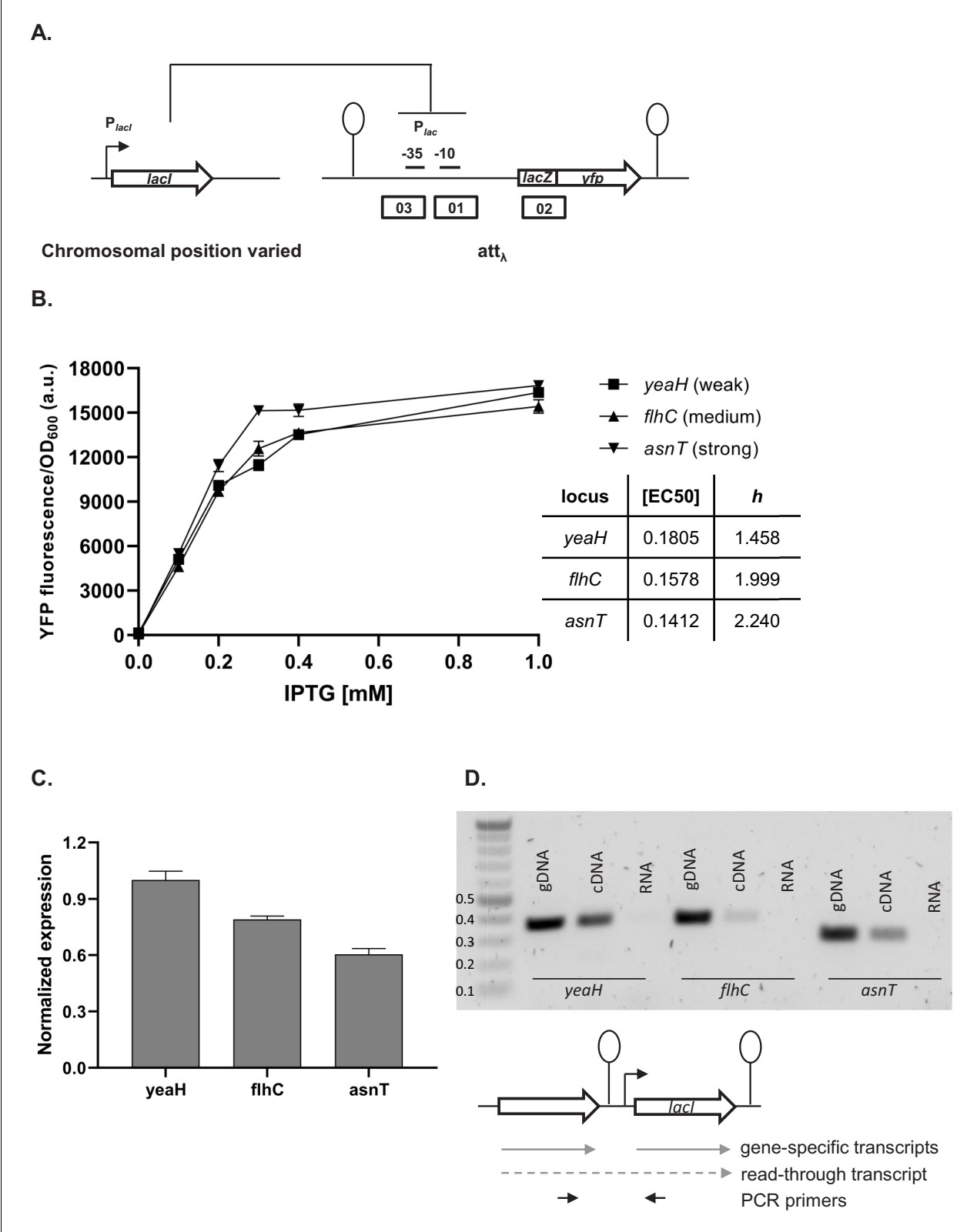

**Figure 6.** Effect of genetic context of *lacI* repressor chromosomal position on P*lac* activity. (**A**) Diagram of interactions between LacI repressor and the promoter it represses, as well as genetic architecture of the DNA fragments integrated into MG1655 Δ*lacI* Δ*lacZYA* strain. Promoters are marked as bent arrows, terminators are represented by vertical bars and a circle, operators as rectangles. (**B**) P*lac* activity in cells carrying *lacI* inserted at different loci after a weak (*yeaH*), medium (*flhC*), and strong (*asnT*) terminator grown in different concentrations of isopropyl β-D-thiogalactopyranoside (IPTG).
*Figure 6 continued on next page*

*Figure 6 continued*

*yfp* levels were measured in exponentially growing cells. Graph shows means for at least three independent biological replicates. (**C**) RT-qPCR quantification of *lacI* transcript in strains described above. RT-qPCR was performed, using *lacI*-specific primers. The induction ratios were calculated relative to the strain with insertion in *yeaH* locus. (**D**) Ethidium bromide-stained 1% agarose gel of PCR products obtained with primers spanning the intergenic region between the upstream gene (*yeaH*, *flhC*, or *asnT*) and *lacI*. Templates for PCR were: chromosomal DNA (gDNA), cDNA as used for RT-qPCR above, and RNA.

The online version of this article includes the following figure supplement(s) for figure 6:

**Figure supplement 1.** P$_{lac}$ activity differs significantly in cells carrying *lacI* inserted after a weak (*yeaH*), medium (*flhC*), and strong (*asnT*) terminator grown in different concentrations of isopropyl β-D-thiogalactopyranoside (IPTG).

thus can qualitatively and quantitatively change the function a GRN performs. Hence, the phenotype of a GRN cannot be fully understood without reference to the local genetic context of its individual network components. We find that our GRN can access multiple phenotypes by simply shuffling the relative order of TUs without any changes in promoters and coding sequence. Thus, changes in regulatory connections between single regulons can be achieved solely by changes in local genetic context, which represent a category of mutations that is to be contrasted from single base pair mutations. The local genetic context is not limited to proximal transcription factors which are part of the same GRN, like our synthetic plasmid system that represents the most direct example. In fact, the local context effects and in particular transcriptional read-through can occur anywhere on the chromosome (*Figure 6*), indicating that any chromosomal rearrangement has the potential to alter not only the expression levels of a gene but importantly also the function of a GRN.

Local genetic context of a TU can change after a deletion, duplication, insertion, inversion, or translocation event (*Periwal and Scaria, 2015*). These mutational events are often mediated by mobile genetic elements, and their rates depend on the type of mobile element, the precise genomic location as well as the organism (*Díaz-Maldonado et al., 2015*; *Periwal and Scaria, 2015*; *Steinrueck and Guet, 2017*). Reported rates span a wide range (from $10^{-3}$ to $10^{-8}$ per cell per generation), but are typically orders of magnitude higher than rates of point mutations (*Hudson et al., 2002*; *Saito et al., 2010*; *Sousa et al., 2013*; *Tomanek et al., 2020*). Given this elevated frequency of small- and large-scale genomic rearrangements in various bacterial species, changes in local genetic context have the potential to shape bacterial phenotypes even in the absence of sequence changes. Specific local genetic contexts could have arisen in response to selection for changes in gene expression levels. Indeed, genomic rearrangements were found to significantly change expression patterns in numerous organisms, including *E. coli*, *Bordetella pertussis*, and *Lactobacillus rhamnosus* (*Brinig et al., 2006*; *Douillard et al., 2016*; *Raeside et al., 2014*; *Weigand et al., 2017*). It needs to be stressed that the impact of local genetic context of GRN elements on fitness will strongly depend on the network's output. Examples of how even a small effect can be strongly amplified further downstream in a regulatory network are the regulatory circuit governing lysogenic and lytic states of phage lambda, or the processes behind entry into sporulation or genetic competence in *Bacillus subtilis* (*Dubnau and Losick, 2006*; *Narula et al., 2012*; *Ptashne, 2004*; *Smits et al., 2005*).

Growing evidence has been emerging that transcriptional read-through is widespread (*Cambray et al., 2013*; *Conway et al., 2014*; *Lalanne et al., 2018*; *Yan et al., 2018*). This raises the question of whether (potentially ubiquitous) the lack of clearly defined gene and operon boundaries can be disruptive, or alternatively if this lack of precise functional boundaries is used for the benefit of the cell. The answer is likely gene-specific, reflecting the complexity of the numerous molecular mechanisms acting at the level of a single gene. On one hand, in order to impact the phenotype, transcriptional read-through into neighboring genes needs to reach an effective threshold, and increases in expression levels can be down-regulated post-transcriptionally (*Lalanne et al., 2018*). On the other hand, fitness benefits have been observed in strains with global changes in Rho-dependent termination levels (*Freddolino et al., 2012*; *Lee and Helmann, 2014*; *Steinrueck and Guet, 2017*; *Tenaillon et al., 2012*), and it is likely that local transcriptional read-through at specific intrinsic terminators is also beneficial under certain conditions. As rates of transcriptional read-through can be condition-dependent (*Yan et al., 2018*), there are many complex molecular interactions left to disentangle regarding the fitness effects, whether beneficial or detrimental, of inefficient transcriptional termination.

We find that transcriptional read-through is an important molecular mechanism behind the effects we observe in our synthetic GRN system. Changes in the strength of transcriptional termination generally require just a small number of mutations, often only individual point mutations (*San Millan et al., 2009*; *Schuster et al., 1994*; *Weigand et al., 2017*). Importantly, despite our synthetic GRN having been designed to restrict transcriptional read-through by using a single very strong transcriptional terminator, we observed a variety of phenotypes our GRN can access (*Cambray et al., 2013*). Naturally occurring transcriptional terminators cover a wide range of efficiencies and hence have potential to create a large number of regulatory connections between neighboring TUs (*Cambray et al., 2013*; *Reynolds et al., 1992*).

Most studies on chromosomal position effects focus explicitly on molecular mechanisms other than transcriptional read-through by insulating a reporter system with strong terminators (*Berger et al., 2016*; *Block et al., 2012*; *Sousa et al., 1997*). Given the fact that endogenous terminators vary widely in their efficiency, we argue that transcriptional read-through from neighboring genes is an inherent component of chromosomal position effects. It can significantly add to other genetic context-dependent effects resulting from gene dosage or DNA supercoiling. Such complex interplay of mechanisms can be seen in our synthetic genetic system, as not all phenotypes we see can be explained by transcriptional read-through alone and there are likely other molecular mechanisms of a more global nature influencing the phenotypes of our GRN. This observation highlights how challenging it is to disentangle all of the complex genetic interactions even in a very simplified synthetic GRN built out of the best-characterized transcription factors. Using simple synthetic systems helps to dissect and understand the dynamics of more intricate and complex cellular interactions, following the tradition of simple model systems that have been powerful throughout the history of molecular biology. Understanding of how synthetic genetic systems behave can be greatly aided by modeling approaches. However, often times the mathematical models can be complex in their implementation and thus difficult to interpret. As we show here, simple, minimalistic models can give important insights into fundamental biological mechanisms. Moreover, transcriptional read-through is a component rarely included in the modeling of GRNs. Yet, its effect can be readily added into any modeling framework.

Gene expression and its regulation are influenced by multiple coexisting molecular mechanisms, through the concerted action of DNA binding proteins, including RNA polymerase, transcription factors, topoisomerases, and nucleoid-associated proteins acting at different levels of organization: from short promoter sequences to mega-base large DNA macro-domains (*Junier, 2014*; *Lagomarsino et al., 2015*). Here, we show that the local genetic context created by the relative TU order can act as one of the genetic mechanisms shaping regulatory connections in regulons (*Figure 7*). Changes in local genetic context have the potential to place an individual TU into two independent regulons without the need to evolve complex regulatory elements. Transcriptional read-through, by enabling a diversity of gene expression profiles to be accessed by shuffling of individual TUs, may be one of the mechanisms shaping the evolutionary dynamics of bacterial genomes. Indeed, the fact that gene expression levels of one gene can be influenced by the gene expression levels of its immediate neighbor has important consequences for the evolution of operons. For a long time it has been debated whether any selective advantage is gained from the physical proximity of two TUs and how this physical proximity can be maintained before common transcription factor-based transcriptional regulation can evolve (*Lawrence and Roth, 1996*). We suggest that physical proximity alone can result in increased co-expression due to transcriptional read-through and thus can be advantageous by changing gene expression patterns without the need for any changes in promoter sequences or to the specificity of transcription factors. Indeed, correlated expression of genes reaching beyond the level of an operon has been recently observed (*Junier et al., 2016*; *Junier and Rivoire, 2016*). Our results also have important implications for comparative genomics, as sequence conservation does not necessarily equal functional conservation. Finally, there is a lesson for engineering living systems, as our results underscore the importance of understanding how nature itself can compute with GRNs (*Guet et al., 2002*; *Kwok, 2010*).

The simple synthetic and endogenous examples of GRNs we studied here show how local genetic context can be a source of phenotypic diversity in GRNs, as the expression of a single gene or operon can be linked to levels and patterns of gene expression of its immediate chromosomal neighborhood. Systematic studies that utilize simple synthetic systems offer the promise of understanding how the genetic elements interact and result in the diversity of phenotypes we observe.

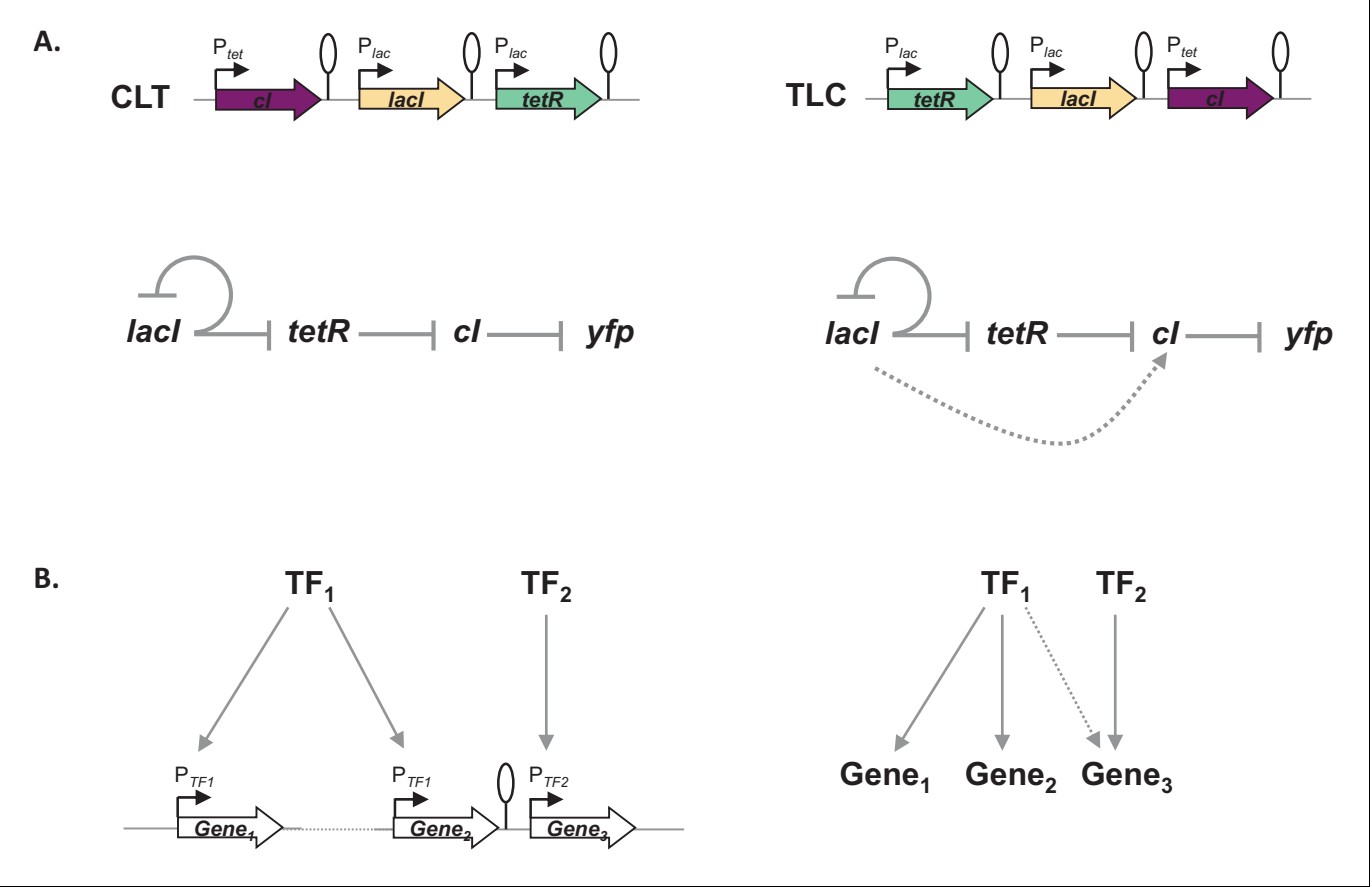

**Figure 7.** Architecture of a regulon depends on local genetic context. (**A**) Diagram of interactions within our gene regulatory network (GRN) in two different transcriptional unit (TU) arrangements: CLT and TLC. (**B**) Regulatory patterns in two regulons with overlapping components. Genes Gene$_1$ and Gene$_2$ are regulated via transcription factor-operator interactions with regulator TF$_1$. Gene Gene$_3$ is regulated by TF$_2$ binding and by TF$_1$ via local genetic context effects. (**A and B**) Solid lines represent interactions between transcription factors and the promoters they control, dashed line represents effects resulting from local genetic context. Promoters are marked as bent arrows, terminators are represented by vertical bars and a circle.

## Materials and methods

### Predicted effects of transcriptional read-through into each of the TFs

In our synthetic GRN, repressor-encoding genes are separated by identical T1 terminators. Transcriptional read-through can in principle happen at any terminator. This scenario is also incorporated in our model. However, in our experimental approach we focused only on read-through into *cl*, which is the network element that directly regulates the level of the measured output *yfp* by binding to its promoter P$_R$ (*Figure 1A*). Transcriptional read-through into *tetR* or *lacI* from *cl* would happen only after induction with aTc, when output is insensitive to the presence of LacI and TetR. Transcriptional read-through from *lacI* into *tetR* can only increase expression of *tetR*, which even without this effect fully represses P$_{tet}$. Transcriptional read-through from *tetR* into *lacI* could potentially make a difference when strains are grown without IPTG, but due to the negative feedback loop and tight repression of P$_{tet}$ these effects are likely too subtle to be visible in our set-up. Thus, for the particular network topology of this study, any changes in levels of LacI and TetR due to transcriptional read-through cannot impact in any way the level of YFP.

### Threshold for assigning a phenotype to individual GRNs

*Figure 1—figure supplement 2* and *Figure 1—figure supplement 3* show fluorescence levels of all the strains carrying different TU order permutations of our network. To define a phenotype that each of the GRNs can achieve, we assign a binary output value for each input state. For each strain,

fluorescence was normalized to the highest expression level at the given time point. The OFF state was defined as at least threefold repression compared to the highest ON state. Moreover, the minimal ON value in each GRN was required to be at least threefold greater than the maximal OFF value. A distribution of logical phenotypes for a varying threshold can be seen in *Figure 1—figure supplement 5*. In the mathematical model, the same procedure is followed. Expression values coming from the mathematical model are normalized by the highest expression value and then a threshold is applied to determine ON and OFF states. The threshold value is constant across all gene orders and orientations.

## Cloning of 48 TU order permutations

Despite repeated attempts, cloning of 11 of the 48 TU order permutations failed. Eight of them were not clonable in either orientation with respect to the plasmid backbone. For three networks, we were able to clone only one orientation (*Figure 1—figure supplement 3B*). Generally, we experienced increased difficulties in cloning GRNs where TUs were not oriented in one direction, which may at least in part be traced back to a number of highly homologous sequences in our plasmids.

## A model for the impact of transcriptional read-through on gene expression

In order to test the mechanistic basis of the changes in gene expression caused by changes in TU order and orientation, we developed a mathematical model that takes into account transcriptional read-through between adjacent TUs. The basic scheme is depicted in *Figure 1—figure supplement 1*. We model the rate of transcript production by a constant term ($k_i$, the constitutive expression rate of the promoter), and an input dependent term that models the repression by other components $\left(-(1-I)\frac{X}{K_X+X}\right)$ and a degradation term (-δ X).

$$\frac{dL}{dt} = k_L - (1-I_1)\frac{L}{K_L+L} - \delta L + r_\chi^L$$
$$\frac{dT}{dt} = k_L - (1-I_1)\frac{L}{K_L+L} - \delta L + r_\chi^T$$
$$\frac{dC}{dt} = k_C - (1-I_2)\frac{L}{K_T+L} - \delta L + r_\chi^C$$

L, T, and C correspond to *lacI*, *tetR*, and *cI*, respectively, $I_1$ and $I_2$ correspond to the presence or absence of IPTG and aTc, respectively. $I_1$ and $I_2$ can take only one of two values, 0 or 1. It should be noted that *lacI* and *tetR* share the same promoter and are therefore controlled by the same rates of production.

Importantly, we include a term $r_\chi$ that models transcriptional read-through. This term takes into account the order and orientation of the specific gene network. When two genes, A and B, are adjacent to each other and share the same orientation this term will take the values $r_{\overrightarrow{AB}}^A = 0$ for gene A and $r_{\overrightarrow{AB}}^A = \mu A'$ for gene B, where A' corresponds to the rate of transcription of gene A $\left(k_A - (1-Ix)\frac{A}{K_A+A}\right)$. The output of the network is an inverse threshold function of the expression level of *cI*, so that the output is ON if *cI* expression is below the threshold $\tau$ and OFF if above it.

In order to obtain the function realized by each of the networks, this system of differential equations is solved for the steady state for the four possible states of the $I_1$ and $I_2$ (0,0), (0,1), (1,0), and (1,1). For each network, the expression levels are normalized by the state with the highest expression and then a threshold is applied to the expression level of *cI* to determine if the network is in an ON or OFF state (see above). For mathematical simplicity and to reduce the number of free parameters we assumed: $k_L = k_C = 1$, $\delta = 1$, $K_L = K_T = K$, leaving essentially two free parameters: the half-repression point K and the read-through rate $\mu$.

In order to obtain the parameters region that allows this system of ordinary differentials equations to fit the experimental data, we performed a grid search in these two parameters. The results for $\tau$ = 2/5 can be seen on *Figure 1—figure supplement 1C*.

## Bacterial strains and growth conditions

All strains used in this study are derivatives of *E. coli* MG1655 and are listed in *Supplementary file 1*. Plasmids are listed in *Supplementary file 3*. Strain and plasmid construction is described in detail below. M9CA+glycerol medium (1× M9 salts, 0.5% glycerol, 2 mM MgSO$_4$, 0.1 mM CaCl$_2$, 0.5%

casamino acids) was routinely used for bacterial growth unless otherwise stated. Selective media contained ampicillin at 100 µg ml$^{-1}$, kanamycin at 50 µg ml$^{-1}$ (for plasmid located resistance cassette) or 25 µg ml$^{-1}$ (for chromosomally located resistance cassette), and chloramphenicol at 15 µg ml$^{-1}$. Solid media additionally contained 1.5% (w/v) agar. IPTG was used at 1 mM unless stated otherwise, aTc at 100 ng ml$^{-1}$.

## Strain and plasmid construction

All strains and plasmids used in this study are listed in *Supplementary file 1* and *Supplementary file 3*, respectively. Strain used for measuring GRN behavior was based on TB201, which is an *E. coli* MG1655 derivative carrying a att$_{P21}$::P$_R$-*yfp* allele. Δ*lacI*785::*kanR* and Δ*lacZ*4787::*rrnB*-3 alleles were transduced (P1) into TB201 from JW0336, and the Kan$^R$ marker removed (pCP20 [*Cherepanov and Wackernagel, 1995*]), resulting in strain **ASE023**. *recA* gene of strain ASE023 was in-frame deleted using λ Red recombination (*Datta et al., 2006*). Kan$^R$ marker was amplified from pKD13 (*Datsenko and Wanner, 2000*) and introduced into the *recA* gene of ASE023. Kan$^R$ cassette was subsequently removed (pCP20 [*Cherepanov and Wackernagel, 1995*]) resulting in strain **KT131**. Strain **KT132** was used for measuring behavior of networks containing *yfp* instead of *cI* and was constructed as described above, with the parent strain being MG1655 instead of TB201. Strain **Frag1B** was used to supply constitutively expressed *tetR* encoded on the chromosome and measure orientation-dependent repression of P$_{LtetO1}$.

Plasmids were constructed by using standard cloning techniques (*Sambrook and Russell, 2001*) with enzymes and buffers from New England Biolabs, according to the respective protocols. All primer sequences used for this study are listed in *Supplementary file 2*. For cloning and plasmid propagation *E. coli* Frag1D was grown routinely in lysogeny broth (LB) at 30℃ with agitation (240 rpm). All plasmids and strains were verified by sequencing.

To facilitate directional cloning of the DNA fragments, and at the same time reduce the background of clones containing empty plasmids, we inserted a DNA fragment encoding *mCherry* flanked with BglI sites into the cloning vector. This facilitated isolation of completely cut vector as well as identification of background clones on a plate due to their fluorescence. The fragment encoding *mCherry* was amplified from vector pBS3Clux (*Radeck et al., 2013*) and cloned into PstI and SmaI sites of vector pLA2 (*Haldimann and Wanner, 2001*), creating plasmid **pAS017**. The plasmid used for cloning of GRN permutations, **pAS019**, was constructed by amplifying the vector backbone (consisting of kanamycin resistance cassette, and SC101* origin of replication) of pZS*2R-*gfp* and inserting the BglI sites-flanked mCherry cassette from pAS017 into its ScaI and EcoRI sites.

Fragments encoding P$_{LlacO1}$-*tetR*, P$_{LlacO1}$-*lacI*, and P$_{LtetO1}$-*cI* were amplified from the original D052 plasmid (*Guet et al., 2002*) and cloned into XhoI and XbaI sites of vector pZS*12-*gfp* (simultaneously removing the *gfp* cassette), resulting in plasmids **pAS014**, **pAS015**, and **pAS016**, respectively. All three repressors are tagged with *ssrA* degradation tag to reduce the half-life of the proteins (*Keiler et al., 1996*).

For construction of the set of gene order permutations, fragments containing P$_{LlacO1}$-*tetR*, P$_{LlacO1}$-*lacI*, and P$_{LtetO1}$-*cI* were amplified from pAS014, pAS015, and pAS016, respectively. The fragment containing P$_{LtetO1}$-*yfp* was amplified from pZS*11-*yfp*. The primers were designed to create BglI restriction sites flanking the genes so that directional and ordered cloning was possible. Equimolar amounts of the fragments were ligated using T4 DNA ligase. The respective trimer was purified from an agarose gel and cloned into BglI sites of pAS019, resulting in plasmids **pN1-54. pKT10**, the empty control plasmid, was constructed by removing *mCherry* from pAS019 using XhoI and SalI and ligating the compatible overhangs. Mutations in the −10 promoter element of plasmids pAS016 and pZS*11-*yfp* were introduced by site-directed mutagenesis. Primer design and mutagenesis were performed according to the manufacturer's instructions for the QuikChange II site-directed mutagenesis kit (Agilent Technologies) resulting in plasmids **pAS023** and **pAS024**. Plasmid pAS023 served as template for construction of network plasmids with P$_{tet-10}$ mutation (**pAS026**, **pAS045-7**, and **pAS050-1**). Terminators T1T2, T*crp*, and T*tonB* were cloned into XbaI site of plasmid pAS015, resulting in plasmids **pAS020**, **pAS021**, and **pAS038**, respectively. These plasmids served as template for construction of network plasmids with exchanged terminators (**pAS039**, **pAS040**, **pAS053**, and **pAS055**). Promoterless *yfp* gene and P$_{LtetO1}$-*yfp* were amplified from pZS*11-*yfp* and cloned with BglI into pAS019 resulting in plasmids **pAS035**, **pAS036**, and **pAS037**. DNA fragments

between repressor genes containing terminators T1, T*crp*, and T*tonB* were cloned into plasmid pAS035, resulting in plasmids **pAS041**, **pAS042**, and **pAS043**, respectively.

Strains with GRNs $P_{LtetO1}$-*cl*-$P_{LlacO1}$-*tetR*-$P_{LlacO1}$-*lacI* (CTL), $P_{LlacO1}$-*lacI*-$P_{LtetO1}$-*cl*-$P_{LlacO1}$-*tetR* (LCT), and $P_{LlacO1}$-*tetR*-$P_{LlacO1}$-*lacI*-$P_{LtetO1}$-*cl* (TLC) integrated into the chromosome were constructed using λ Red recombination (*Datta et al., 2006*). Appropriate fragments including Kan$^R$ cassette were amplified from plasmids pN2, pN3, and pN6 and integrated into phage HK022 attachment site of ASE023 resulting in strains **ASE031**, **ASE032**, and **ASE033**, respectively. Strain with three repressors integrated into separate loci on the chromosome originated from strain KT131, which was subsequently transformed with plasmids **pKT12** (carrying $P_{LlacO1}$-*lacI* and integrating into phage HK022 attachment site), and **pAS022** (carrying $P_{LlacO1}$-*tetR* and integrating into phage λ attachment site). Both pKT12 and pAS022 are based on modified CRIM plasmids (*Haldimann and Wanner, 2001*; *Pleška et al., 2016*). After each round of transformation, the Cam$^R$ marker was removed (pCP20 [*Cherepanov and Wackernagel, 1995*]). The $P_{LtetO1}$-*cl* fragment was integrated into *old* locus using λ Red recombination (*Datta et al., 2006*), resulting in strain **ASE030**.

Strain with $P_{lac(-131-410)}$-*yfp* originated from HG105, which is an *E. coli* MG1655 derivative carrying a Δ*lacZYA* Δ*lacI* allele (*Garcia et al., 2011*). HG105 was transformed with plasmid **pCC01** (carrying $P_{lac(-131-410)}$-*yfp* and integrating into phage λ attachment site) resulting in strain **ASE039**. pCC01 is based on modified CRIM plasmid (*Haldimann and Wanner, 2001*; *Pleška et al., 2016*). Cam$^R$ marker was subsequently removed (pCP20 [*Cherepanov and Wackernagel, 1995*]). To facilitate transduction of *lacI* gene, Cam$^R$ cassette was integrated downstream of *lacI* in strain MG1655 using λ Red recombination (*Datta et al., 2006*), resulting in strain **ASE041**. *lacI* gene and Cam$^R$ cassette were then integrated into *flhC*, *yeaH*, and *asnT* loci, followed by Cam$^R$ cassette removal, resulting in strains **ASE046**, **ASE047**, and **ASE048**, respectively.

## Fluorescence assays

YFP fluorescence of *E. coli* strains harboring different permutational GRN variants was assayed using a Synergy H1 microplate reader (BioTek). Strains were grown in a 96-well plate at 30°C with aeration on a microplate shaker in the dark. Routinely, M9CA+glycerol medium was used. Strains ASE031, ASE032, and ASE033 were grown in LB. Overnight cultures started from single colonies were diluted 1:1000 into fresh medium (supplemented with aTc and/or IPTG as indicated) and grown to reach exponential phase. $OD_{600}$ and fluorescence (excitation 515, emission 545; endpoint-reads; gain 90; emission side: bottom) were recorded. For strains grown in LB, cells were centrifuged and resuspended in PBS (supplemented with 1 mM $MgSO_4$ and 0.1 $CaCl_2$[*Tomasek et al., 2018*]) prior to measurements. Specific fluorescence activity is given by the raw fluorescence output normalized by cell density. For GRN permutations fluorescence is reported normalized to the fully unrepressed $P_R$ promoter.

## Total RNA purification

Strains were inoculated from single colonies and grown in 10 ml M9CA+glycerol medium for 8 hr with aeration in the dark. Total RNA was extracted from approximately $5 \times 10^8$ cells using RNAprotect Bacteria reagent and RNeasy Mini kit (Qiagen). Briefly, after removal of the growth medium the cells were resuspended in 500 µl M9CA+glycerol medium and two volumes of RNAprotect Bacteria reagent. Thereafter cells were enzymatically lysed and digested with lysozyme and Proteinase K according to the manufacturer's recommendation.

The extracted total RNA was purified from residual plasmid DNA using the DNA-*free* DNA removal kit (Thermo Fisher Scientific) using 4 U rDNase at 37°C for 1 hr in total. First two U rDNAse were added and after 30 min two more units were added for another 30 min incubation time. The RNA concentration was measured using the NanoDrop 200 UV-Vis spectrophotometer (NanoDrop products, Wilmington, DE) and the integrity of the purified RNA was verified on an agarose gel. RNA purity was verified using 1× One*Taq* 2× master mix and 0.2 µM primers KTp38 and KTp39 for plasmid networks and primers AS271-2, AS277, and AS280 for chromosomal *lacI* strains running an end-point PCR.

## cDNA preparation and quantitative real-time PCR

cDNA was reverse transcribed using the iScript cDNA synthesis kit (BioRad) supplemented with random hexamers. 1 μg total RNA was used as template in a 20 μl reaction yielding approximately 50 ng/μl cDNA. As no reverse transcriptase control the reverse transcriptase reaction was performed with all components except the reverse transcriptase to verify the absence of DNA contaminations. The products of the reverse transcriptase reaction were column purified.

Measurement of transcript abundance was performed by quantitative real-time RT-PCR using the GoTaq qPCR Master Mix (Promega, Mannheim, D) supplemented with SYBR Green according to the manufacturer's procedure with minor modifications. 100 pg cDNA for plasmid networks and 500 pg of cDNA for chromosomal strains were used. Primer pairs (*Supplementary file 2*) were designed to quantify the transcription level of *cI* and *lacI*. Expression of the kanamycin resistance marker on the network plasmids and *cysG* on the chromosome were monitored as constitutive references. The qPCR reaction was carried out on the BIO-RAD qPCR C1000 system using 0.3 μM of the respective primers for *cI* and *kanR* amplification and 0.3 μM and 0.9 μM of the respective primers for *lacI* and *cysG* amplification at an annealing temperature of 62℃. The amplification efficiency, the linearity, including the slope and the $R^2$, and specificity of each primer pair were determined by amplifying experimental triplicates of a serially dilution mixture of pN1 plasmid or genomic DNA of one of the chromosomal strains (1 ng to 1 pg). Using the conditions mentioned above the amplification efficiency was almost equal for all primer pairs. Expression of *cI* and *lacI* was calculated as fold changes using the comparative $C_T$ method ($\Delta\Delta C_T$) (*Livak and Schmittgen, 2001*).

## Northern blot assay

After overnight ethanol precipitation (0.1 vol 1 M sodium acetate, 2.5 vol ethanol) 10 μg of total RNA was denatured with 2× RNA loading buffer (4% 10× TBE-DEPC, 0.02% xylene cyanol, 0.02% bromophenol blue, 94% formamide) for 15 min at 65℃. The high range RNA molecular weight marker (RiboRuler High Range RNA Ladder, Thermo Fisher Scientific) was dephosphorylated with FastAP Thermosensitive Alkaline Phosphatase (Thermo Fisher Scientific) and 5'-end labeled with γ−32P-ATP and T4 Polynucleotide Kinase (Thermo Fisher Scientific) according to the manufacturer's instructions. The denatured RNA samples and high range RNA molecular weight marker were separated on a 1.2% denaturing agarose gel (1.2× MOPS, 19.4% formaldehyde; running buffer: 1× MOPS, 16.2% formaldehyde) for 2 hr. After rinsing the gel three times with DEPC water the RNA was blotted onto a Nylon membrane (Hybond-N+, GE Healthcare, Amersham, UK) overnight using 20× SSC (3 M NaCl, 0.3 M sodium citrate). The membrane was exposed to UV light, rinsed with 6× SSC, and again exposed to UV before pre-hybridizing for 2 hr at 55℃. The pre-hybridization solution contains 0.04% BSA, 0.04% PVP, 0.04% Ficoll, 10 mM EDTA, 5× SSC, 0.2% SDS, 0.1% dextran, and 0.1 mg ml$^{-1}$ salmon sperm DNA. Hybridization was performed overnight at 55℃ using 2 pmol of *cI* and *lacI* specific 32P 5'-end labeled oligonucleotides, respectively. The probes were labeled using γ−32P-ATP and 10 U T7 polynucleotide kinase (Thermo Fisher Scientific) for 1 hr at 37℃ and column purified afterwards. The hybridization solution was identical to the pre-hybridization solution but without dextran and salmon sperm DNA. After rinsing two times with 0.5× SSC and washing 20 min with 1× SSC + 0.1% SDS and 15 min with 0.5× SSC + 0.1% SDS, the membrane was dried for 30 min at room temperature. The blot was exposed up to several days, and the hybridization signals were detected using Phosphorimager from Molecular Dynamics. As reference signal for normalization the kanamycin resistant gene on the network plasmids was used.

## FACS

Flow cytometry was performed on a FACSCanto II analyzer (BD Biosciences, San Jose, CA). Sensitivity of the lasers was determined within the daily setup using BD FACS 7-color setup beads. For scatter detection the 488 nm laser was used: the forward scatter (FSC) detector was set to 560 V, and the side scatter (SSC) detector was set to 374 V. Both signals were collected through a 488/10 nm band-pass filter. Cells were plotted on a log scale with thresholding on FSC and SSC at 1000. The green emission from the FITC-H channel was collected through a 530/30 nm band-pass filter using 488 nm laser and the detector was set to 473 V. The fluorescence signal observed from a physiologically distinct subpopulation, gated on FSC-H and SSC-H, was biexponentially transformed. Cells were grown as for fluorescence population measurements, and after 6 hr of growth 15 μl

aliquots was frozen overnight adding 15 µl 30% glycerol in M9 buffer (1× M9 salts with Ca/Mg). After thawing, the samples were diluted in cold M9 buffer to reach an event rate of approximately 500 events/s at medium flow rate. 20,000 events were recorded using high throughput sampler (HTS). The mean fluorescence of approximately 10,000 gated cells similar in size and shape (FSC-H) and cellular complexity (SSC-H) was determined. Events were gated and values were extracted using FlowJo software (version 10.0.7, FlowJo LLC, Tree Star).

## Statistical analysis

To analyze *yfp* fluorescence data measured to assess whether genomic location affected response to IPTG, we conducted an analysis of variance (ANOVA) at each concentration separately, with the measured fluorescence as the response factor and the genomic concentration as the fixed factor (*Supplementary file 4*). We followed up these with a series of Tukey's multiple comparisons tests performed separately for each IPTG concentration, to directly compare the expression levels between genomic locations (*Figure 6—figure supplement 1*).

## Acknowledgements

We thank J Bollback, L Hurst, M Lagator, C Nizak, O Rivoire, M Savageau, G Tkacik, and B Vicozo for helpful discussions; A Dolinar and A Greshnova for technical assistance; T Bollenbach for supplying the strain JW0336; C Rusnac, and members of the Guet lab for comments. The research leading to these results has received funding from the People Programme (Marie Curie Actions) of the European Union's Seventh Framework Programme (FP7/2007-2013) under REA grant agreement n˚ 628377 (ANS) and an Austrian Science Fund (FWF) grant n˚ I 3901-B32 (CCG).

## Additional information

### Funding

| Funder | Grant reference number | Author |
|---|---|---|
| FP7 People: Marie-Curie Actions | 628377 | Anna Nagy-Staron |
| Austrian Science Fund | 3901-B32 | Calin C Guet |

The funders had no role in study design, data collection and interpretation, or the decision to submit the work for publication.

### Author contributions

Anna Nagy-Staron, Conceptualization, Data curation, Supervision, Funding acquisition, Validation, Investigation, Visualization, Methodology, Writing - original draft, Project administration, Writing - review and editing; Kathrin Tomasek, Data curation, Validation, Investigation, Visualization, Methodology, Writing - original draft, Writing - review and editing; Caroline Caruso Carter, Investigation; Elisabeth Sonnleitner, Resources, Investigation; Bor Kavčič, Data curation, Visualization; Tiago Paixão, Data curation, Formal analysis, Validation, Investigation, Visualization, Methodology, Writing - original draft, Writing - review and editing; Calin C Guet, Conceptualization, Resources, Supervision, Funding acquisition, Project administration, Writing - review and editing

### Author ORCIDs

Anna Nagy-Staron 
Kathrin Tomasek 
Caroline Caruso Carter 
Bor Kavčič 
Calin C Guet 

### Decision letter and Author response

Decision letter https://doi.org/10.7554/eLife.65993.sa1

Author response https://doi.org/10.7554/eLife.65993.sa2

---

## Additional files

### Supplementary files

- Supplementary file 1. Strains used in this study.
- Supplementary file 2. Oligonucleotides used in this study.
- Supplementary file 3. Plasmids used in this study.
- Supplementary file 4. ANOVA test statistics.
- Transparent reporting form

### Data availability

Plasmid sequences are provided in IST Research Depository, https://doi.org/10.15479/AT:ISTA:8951.

The following dataset was generated:

| Author(s) | Year | Dataset title | Dataset URL | Database and Identifier |
|---|---|---|---|---|
| Nagy-Staron AA | 2020 | Sequences of gene regulatory network permutations for the article "Local genetic context shapes the function of a gene regulatory network | https://research-explorer.app.ist.ac.at/record/8951 | IST Research Depository, 10.15479/AT:ISTA:8951 |

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
