## [Decision Letter]

**Acceptance summary:**

We really enjoyed this insight into the complexities of even simply gene regulatory networks, which you show are nowhere near as simple as we thought. Indeed, We think insights into the effects of transcriptional read-through will interest many concerned with the connection between genotype and phenotype.

**Decision letter after peer review:**

[Editors’ note: the authors submitted for reconsideration following the decision after peer review. What follows is the decision letter after the first round of review.]

Thank you for submitting your work entitled "Local genetic context shapes the function of a gene regulatory network" for consideration by *eLife*. Your article has been reviewed by three peer reviewers, and the evaluation has been overseen by a Reviewing Editor and a Senior Editor. The reviewers have opted to remain anonymous.

Our decision has been reached after considerable consultation between the reviewers. Based on these discussions and the individual reviews below, we regret to inform you that your work will not be considered further for publication in *eLife*.

It was not easy coming to this decision. All reviewers and the reviewing editor found much merit in the work. Particularly appreciated was the extensive generation of gene combinations and evidence of effects on the GRN. But the agreed position among the reviewers that more was required in order to rise about the bar necessary for publication in *eLife*. In particular, all felt that lack of insight into the molecular basis of the observed effects amounted to a significant shortcoming.

Recognising that this is something that could be overcome there was discussion as to whether this could be achieved within a two-month window. Of course all recognised that in the time of covid this is an unrealistic time frame, but nonetheless, it was felt that significant additional work was required. This said, we would be happy to consider a fresh resubmission at some future time if the authors were able to deliver additional information as to the effect of the GRN on transcriptional read-through.

Reviewer #1:

I flipped about with this manuscript, from mildly enthusiastic to very enthusiastic; back and forth. On balance though, I think it provides useful additional understanding of factors affecting the evolution of GRN. Like amplification, and various structural changes, those affecting the relative placement of genes can have unrecognised effects.

The work itself is thorough and clean. A very simple contrived construct is used and the various permutations of possible gene placements are comprehensively analysed. The work in chromosomal context is very nice, and especially so the work placing the three genes in separate chromosomal locations.

Where I was left hanging was with regard to the molecular mechanism. I buy the data showing that the mechanism involves transcriptional read-through, but the reader is left with no additional understanding. The obvious unaddressed question is why does transcriptional read through occur given otherwise identical terminators?

Reviewer #2:

The authors describe an issue well-known to synthetic biologist who are trying to build synthetic gene circuits in a predictive manner, namely context-dependent effects. More specifically, they are focusing on the order of the individual transcriptional units of a circuit and observe different behaviours, not only quantitatively, but also qualitatively. I like the approach of using a synthetic circuit, I think it is interesting to quantify such an effect in more detail and also to put it into an evolutionary context. However, I did not enjoy the manuscript as much as I was hoping for after reading the Abstract, due to following reasons:

1) Very little thought was put into making the figures self-explanatory. I had to go several times for and back between text, figure and figure captions to understand what is actually shown, how the experimental results compare to the expectations and how the results compare across the different systems. Some extra schemes might help here. This applies to all of the figures, but I will explain it in more detail for Figure 1 and Figure 4

Figure 1A: Explain and show in detail what phenotype is expected. Maybe use the mathematical model used later to illustrate the expression level of each involved protein under all conditions. As the networks involve a negative feedback loop, I find this non-trivial. The expected behaviour will depend on the repression strength of *lacI*. B. also label the prompters. C: instead of only the labels "CfLfTf" etc, add the small schemes, including labelling.

Figure 4A: Add the interactions into the scheme and the expected behaviours for each case (with/without readthrough). Also include YFP in this scheme. This might be trivial for the authors, but it is not for a reader, who has not worked years on that project

The figures would also be easier to read if you used some colours.

2) The experimental evidence for the author's conclusion that transcriptional read-through is the main factor for some of the unexpected phenotypes is rather slim. The only evidence is Figure 4. I would like to see more evidence. They say a RNAse site prevents it to detect in Western blotting. But as the system is synthetic, it is very simple to remove that site. Moreover, they should insert about 10 different terminators and use qPCR or RNAseq to quantify the read-through in order to get an idea how wide-spread this phenomenon is. It would also be easy to put multiple terminators in a row and see if that recovers the expected behaviour. The beauty of a synthetic system is that those experiment are done in few weeks. (Of course not if the lab is shut down due to COVID-19…)

We observed that it is very easy to get unwanted promoter in sequences. That is supported by literature that shows that is very easy to get promoters from random sequences. Could this be an alternative explanation? Please discuss whether and how this can be distinguished from read-through?

3) The effect in the native circuit is rather modest. Is this even significant? No statistical test was performed. Again, the read-through should be quantified by qPCR or RNAseq.

Reviewer #3:

Overall, there are some conceptual issues with the study, as presented. The study does demonstrate physical context of genes does impact outputs. However, there are some issues.

The research and findings are not well placed in the field. There are a number of notable studies that have examined physical location of genes and their impact on regulatory networks and/or physiological outputs using natural and synthetic systems. For example, phage T7 was refactored and these synthetic T7 phages have notable differences in infectivity, etc. from their WT T7 counterpart (Chan, 2005). Additionally, a study by Wu and Rao (2010) has examined the impacts of genetic arrangement on the outputs of autoregulatory circuits. There are several other similar studies, beyond the few cited in this manuscript. How this approach builds on these studies is not made very clear.

The study seems to have honed in on one particular aspect that may affect a regulatory network – transcriptional readthrough. This seems to be explanatory for this system, and is likely a direct use of using very strong promoters (P_L derivatives). Use of such strong promoters, regardless of the efficiency of a terminator, will result in significant amounts of readthrough. How transcriptional readthrough generally contributes to regulatory networks is debatable. The data presented to support the importance of readthrough (Figure 5), while possibly statistically significant, does not provide convincing evidence for strong effects. Building on this point, how frequently are transcription factors in a multicomponent network encoded in close proximity? Would transcriptional readthrough provide any explanation as to the broader patterns we see in genome structure – even for *E. coli*?

There is very little explanation for why readthrough into cI has such strong effects, when readthrough into *lacI* or *tetR* have smaller (or no) effects on network behavior. Are these effects on cI strongest because CI is most proximal to your output (YFP production)? Do these effects correlate in any way to DNA binding affinity of each transcription factor?

---

## [Author Response]

[Editors’ note: the authors resubmitted a revised version of the paper for consideration. What follows is the authors’ response to the first round of review.]

Reviewer #1:I flipped about with this manuscript, from mildly enthusiastic to very enthusiastic; back and forth. On balance though, I think it provides useful additional understanding of factors affecting the evolution of GRN. Like amplification, and various structural changes, those affecting the relative placement of genes can have unrecognised effects.The work itself is thorough and clean. A very simple contrived construct is used and the various permutations of possible gene placements are comprehensively analysed. The work in chromosomal context is very nice, and especially so the work placing the three genes in separate chromosomal locations.Where I was left hanging was with regard to the molecular mechanism. I buy the data showing that the mechanism involves transcriptional read-through, but the reader is left with no additional understanding. The obvious unaddressed question is why does transcriptional read through occur given otherwise identical terminators?

We thank the reviewer for the encouraging words and appreciation of this work. We also thank the reviewer for having pushed us to dig deeper in to the mechanistic aspects definitely improved our work. We hope that in the revised version we managed to address the criticism such that the level of enthusiasm is “very” across the entire manuscript.

Specifically, the extensive new experimental data we provide (strains with promoter mutations or exchanged terminator strength using different naturally occuring terminators; qPCR to verify expression levels, Northern blot to visualize read-through transcripts) addresses some of the conundrum related to molecular mechanism. We have also clarified why transcriptional read-through, although likely happening at all terminators in our GRN, has a measurable effect only if it increases expression levels of only one repressor, namely λ CI.

Reviewer #2:The authors describe an issue well-known to synthetic biologist who are trying to build synthetic gene circuits in a predictive manner, namely context-dependent effects. More specifically, they are focusing on the order of the individual transcriptional units of a circuit and observe different behaviours, not only quantitatively, but also qualitatively. I like the approach of using a synthetic circuit, I think it is interesting to quantify such an effect in more detail and also to put it into an evolutionary context. However, I did not enjoy the manuscript as much as I was hoping for after reading the Abstract, due to following reasons:1) Very little thought was put into making the figures self-explanatory. I had to go several times for and back between text, figure and figure captions to understand what is actually shown, how the experimental results compare to the expectations and how the results compare across the different systems. Some extra schemes might help here. This applies to all of the figures, but I will explain it in more detail for Figure 1 and Figure 4.

We apologize for the lack of clarity. We only realized in retrospect how heavy the presentation was. We therefore made a number of changes to all figures, most importantly adding color coding which helps to connect the repressors with conditions in which they change their promoter affinity. We have simplified the labelling of networks, with a capital letter (e.g. C) signifying a gene in forward orientation, and a capital letter with ‘r’ (e.g. C_r_) signifying a gene in reverse orientation, making the figures easier to read. We also added – as suggested – schemes where applicable. Former Figure S1, depicting all permutations of our GRN has been split into two easier to read Figures 1—figure supplement 2 and 3.

Figure 1A: Explain and show in detail what phenotype is expected. Maybe use the mathematical model used later to illustrate the expression level of each involved protein under all conditions. As the networks involve a negative feedback loop, I find this non-trivial. The expected behaviour will depend on the repression strength of lacI. B. also label the prompters. C: instead of only the labels "CfLfTf" etc, add the small schemes, including labelling.

We thank the reviewer for these suggestions – predicted behavior of the network is now illustrated in Figure 1A. We labeled the prompters in Figure 1B by adding a legend as well. Genetic architecture and labelling was added in Figure 1C (as well as in other relevant figures).

Figure 4A: Add the interactions into the scheme and the expected behaviours for each case (with/without readthrough). Also include YFP in this scheme. This might be trivial for the authors, but it is not for a reader, who has not worked years on that project.

We agree with the reviewer that the figure benefits from more explanation. We added interactions within the GRN, including YFP. We also added a short explanation of expected results to the figure caption: “We expected that if GRN behavior is due to transcriptional read-through, mutating the P*_tet_* promoter should not affect the responsiveness to IPTG. If, on the other hand, expression of *cI* was driven only by P*_tet_*, mutating the -10 promoter element should lead to an ALL ON phenotype. Lack of repression in strains with P*_tet-10_* variant after aTc induction further confirms that this promoter variant is inactive.”

The figures would also be easier to read if you used some colours.

Thanks for the suggestion on using colors. We used a consistent color scheme in majority of the figures.

2) The experimental evidence for the author's conclusion that transcriptional read-through is the main factor for some of the unexpected phenotypes is rather slim. The only evidence is Figure 4. I would like to see more evidence. They say a RNAse site prevents it to detect in Western blotting. But as the system is synthetic, it is very simple to remove that site. Moreover, they should insert about 10 different terminators and use qPCR or RNAseq to quantify the read-through in order to get an idea how wide-spread this phenomenon is. It would also be easy to put multiple terminators in a row and see if that recovers the expected behaviour. The beauty of a synthetic system is that those experiment are done in few weeks. (Of course not if the lab is shut down due to COVID-19)

We added a significant amount of additional experimental data to substantiate our conclusions:

a) We constructed a number of strains with *tet* promoter mutations (Figure 4—figure supplement 1)

b) We changed terminator strength by changing T1 terminator to either: stronger double terminator T1T2, weaker T*crp* or T*tonB*, or a combined T1-T*crp* terminator (Figure 5)

c) We also visualized read-through transcripts by Northern blotting (Figure 5—figure supplement 1).

The RNase site within T1 cannot be removed without influencing terminator strength, as it is located within the hairpin. Therefore, as suggested by the reviewer, we exchanged terminators and inserted two stronger double terminators (T1T2 and T1-T*crp*) and two weaker ones (T*crp* or T*tonB*). The strongest terminator recovers the expected phenotype, the second double terminator recovers it only partially. Weaker terminators lead to more transcriptional read-through. We also visualized read-through transcripts using Northern blot.

We find that it is beyond the scope of this paper to demonstrate how widespread transcriptional read-through is especially since this is still a hard and open problem both bioinformatically and experimentally. We are currently preparing a manuscript reporting on measurements of termination efficiency of over 150 *E. coli* and *S. enterica* terminators, which in majority of the cases are weaker than T1 we used in this study. A paper by Cambray et al., 2013, shows that termination efficiencies cover a large range, and that terminators in *E. coli* are in general not 100% efficient in stopping transcription.

We observed that it is very easy to get unwanted promoter in sequences. That is supported by literature that shows that is very easy to get promoters from random sequences. Could this be an alternative explanation? Please discuss whether and how this can be distinguished from read-through?

This is a very valid point. To rule out that there is spurious transcription emerging from a DNA fragment between repressor-encoding genes, we cloned these fragments in front of a promoterless *yfp* gene, but detected no fluorescence (Figure 5—figure supplement 2). Moreover, Northern blots show full length readthrough transcripts. Another important point is that read-through is dependent on IPTG concentration (in all networks showing NOR phenotype, for IPTG gradient see Figure 3D). With all this evidence, we are confident that the effects we see do not come from cryptic promoters.

3) The effect in the native circuit is rather modest. Is this even significant? No statistical test was performed. Again, the read-through should be quantified by qPCR or RNAseq.

We have conducted ANOVA to compare the effect of genomic localization on Plac activity (Results section and Figure 6—figure supplement 1). Transcriptional read-through was quantified by qPCR (Figure 6C) and demonstrated presence of the read-through transcript by end-point PCR (Figure 6D).

Reviewer #3:Overall, there are some conceptual issues with the study, as presented. The study does demonstrate physical context of genes does impact outputs. However, there are some issues.The research and findings are not well placed in the field. There are a number of notable studies that have examined physical location of genes and their impact on regulatory networks and/or physiological outputs using natural and synthetic systems. For example, phage T7 was refactored and these synthetic T7 phages have notable differences in infectivity, etc. from their WT T7 counterpart (Chan, 2005). Additionally, a study by Wu and Rao (2010) has examined the impacts of genetic arrangement on the outputs of autoregulatory circuits. There are several other similar studies, beyond the few cited in this manuscript. How this approach builds on these studies is not made very clear.

We thank the reviewer for these suggestions. Indeed we failed to more properly place the study in a larger context. We thank the reviewer for the suggested references which we used in addition to few others as we revised the Introduction accordingly to better address the literature on context in general, how it affects GRNs, and why GRNs are considered usually to be determined mainly by topology.

The study seems to have honed in on one particular aspect that may affect a regulatory network – transcriptional readthrough. This seems to be explanatory for this system, and is likely a direct use of using very strong promoters (P_L derivatives). Use of such strong promoters, regardless of the efficiency of a terminator, will result in significant amounts of readthrough. How transcriptional readthrough generally contributes to regulatory networks is debatable. The data presented to support the importance of readthrough (Figure 5), while possibly statistically significant, does not provide convincing evidence for strong effects. Building on this point, how frequently are transcription factors in a multicomponent network encoded in close proximity? Would transcriptional readthrough provide any explanation as to the broader patterns we see in genome structure – even for *E. coli*?

These are important and valid points, thank you for raising them. The

P*lac* variant we use is indeed a strong promoter, but so is the terminator T1, which is one of the strongest terminators in *E. coli*. Generally, while promoters are indeed not as strong as the PL derived P*lac*, more importantly the terminators are far less efficient across the genome, hence read-through is a common issue. A high-resolution transcriptome map of *E. coli* shows the extent of transcriptional read-through, with e.g. 75% of convergent operons showing transcription into an adjacent operon (Conway et al., 2014). Also, correlated expression of genes reaching beyond the level of an operon has been observed, suggesting that at least some broader patterns can be explained by transcriptional read-through or rather, transcriptional read-through is likely one of the many factors acting on genomic patterns in a combinatorial fashion (Junier, 2014, Junier and Rivoire, 2016). Bioinformatic studies are very difficult to perform, as predicting terminator strength is still very unreliable.

We would like to point out that we do not claim that transcriptional readthrough leads to *strong* effects (see Discussion: “It needs to be stressed that the impact of local genetic context of GRN elements on fitness will strongly depend on the network’s output. Examples of how even a small effect can be strongly amplified further downstream in a regulatory network are the regulatory circuit governing lysogenic and lytic states of phage λ, or the processes behind entry into sporulation or genetic competence in *Bacillus subtilis*”).

Please note that our conclusions are not limited to multicomponent networks, but we are rather showing that transcriptional read-through can lead to emergence of additional links between nodes in GRN (Figure 7A, B). We rephrased the Discussion of this point in order to be clearer: “The local genetic context is not limited to proximal transcription factors which are part of the same GRN, like our synthetic plasmid system that represents the most direct example. In fact, the local context effects and in particular transcriptional read-through can occur anywhere on the chromosome (Figure 6), indicating that any chromosomal rearrangement has the potential to alter not only the expression levels of a gene but importantly also the function of a GRN."

There is very little explanation for why readthrough into cI has such strong effects, when readthrough into lacI or tetR have smaller (or no) effects on network behavior. Are these effects on cI strongest because CI is most proximal to your output (YFP production)? Do these effects correlate in any way to DNA binding affinity of each transcription factor?

We have added a more detailed clarification for why transcriptional readthrough, although likely happening at all terminators in our GRN, has a measurable effect only if it increases expression levels of only one repressor, namely λ cI.